# Disentangling astroglial physiology with a realistic cell model in silico

Leonid P. Savtchenko[1], Lucie Bard[1], Thomas P. Jensen [1], James P. Reynolds[1], Igor Kraev[2], Nikolay Medvedev[2], Michael G. Stewart[2], Christian Henneberger [1,3,4] & Dmitri A. Rusakov [1]

Electrically non-excitable astroglia take up neurotransmitters, buffer extracellular $K^+$ and generate $Ca^{2+}$ signals that release molecular regulators of neural circuitry. The underlying machinery remains enigmatic, mainly because the sponge-like astrocyte morphology has been difficult to access experimentally or explore theoretically. Here, we systematically incorporate multi-scale, tri-dimensional astroglial architecture into a realistic multi-compartmental cell model, which we constrain by empirical tests and integrate into the NEURON computational biophysical environment. This approach is implemented as a flexible astrocyte-model builder ASTRO. As a proof-of-concept, we explore an in silico astrocyte to evaluate basic cell physiology features inaccessible experimentally. Our simulations suggest that currents generated by glutamate transporters or $K^+$ channels have negligible distant effects on membrane voltage and that individual astrocytes can successfully handle extracellular $K^+$ hotspots. We show how intracellular $Ca^{2+}$ buffers affect $Ca^{2+}$ waves and why the classical $Ca^{2+}$ sparks-and-puffs mechanism is theoretically compatible with common read-outs of astroglial $Ca^{2+}$ imaging.

[1] UCL Institute of Neurology, University College London, London WC1N 3BG, UK. [2] The Open University, Milton Keynes, MK7 6AA, UK. [3] German Center of Neurodegenerative Diseases (DZNE), Bonn 53127, Germany. [4] Institute of Cellular Neurosciences, University of Bonn Medical School, Bonn 53127, Germany. These authors contributed equally: Leonid P. Savtchenko, Lucie Bard. Correspondence and requests for materials should be addressed to L.P.S. (email: skaalsa@ucl.ac.uk) or to D.A.R. (email: d.rusakov@ucl.ac.uk)

Astroglia have emerged as an essential contributor to neural circuit signalling in the brain. In addition to the well-established mechanisms of neurotransmitter uptake and extracellular $K^+$ buffering, electrically passive astrocytes appear competent in handling physiological signals using intracellular $Ca^{2+}$ signals[1–3] that display a variety of dynamic ranges and time scales (reviewed in refs. [4,5]). Tri-dimensional (3D) reconstructions of astroglia using electron microscopy (EM) have long revealed a system of nanoscopic processes[6,7] that pervade the entire cell expanse[8,9]. Deciphering cellular mechanisms that shape $Ca^{2+}$-dependent signalling and physiological membrane currents in this sponge-like system has been a challenge.

In contrast, cellular machineries underpinning neuronal physiology have been understood in great detail. This is partly because it has been possible to interpret electrophysiological and imaging observations in neurons using realistic biophysical cell models, such as those developed in the NEURON environment[10,11]. There have also been numerous attempts to simulate astroglial function, mainly from a reductionist standpoint (reviewed in refs. [12,13]). Aimed at a specific question, such models would normally focus on kinetic reactions inside astroglia[14,15], between astroglial and neuronal compartments[16,17] or on astroglial influences in neuronal networks[18,19]. These studies have provided some important insights into the biophysical basis of astroglial physiology. However, their scope would normally exclude complex cell morphology, intracellular heterogeneities or the impact of $Ca^{2+}$ buffering mechanisms on $Ca^{2+}$ signal readout. Thus, integrating cellular functions of an astrocyte on multiple levels, in one realistic entity in silico, remains to be achieved.

Our aim was therefore three-fold. Firstly, to develop a modelling approach that would recapitulate fine astroglial morphology while retaining full capabilities of biophysical simulations enabled by NEURON. We have therefore generated (MATLAB- and NEURON-based) algorithms and software that (a) use experimental data to recreate the space-filling architecture of astroglia, and (b) make this cell architecture NEURON-compatible. Our case study focused on the common type of hippocampal protoplasmic astroglia in area CA1, which has been amongst the main subjects of studies into synaptic plasticity and neuron-glia interactions[20–22]. We have combined patch-clamp electrophysiology, two-photon excitation (2PE) imaging and 2PE spot-uncaging, fluorescence recovery from photobleaching (FRAP), astroglia-targeted viral transduction $Ca^{2+}$ indicators in vivo, and quantitative correlational 3D EM to systematically document the multi-scale morphology and key physiological traits of these cells. Based on these empirical constrains, we have built a multi-compartment 3D cell model fully integrated into the NEURON environment. The latter was equipped with additional functionalities relevant to astroglia, such as control of tissue volume filling and surface-to-volume ratios, options for extracellular glutamate application and $K^+$ rises, endfoot and gap junctions menus, choice of fluorescence imaging conditions, etc.

Our second objective was to implement this approach as a flexible simulation instrument—cell model builder—capable of recreating and probing various types of astroglia in silico. Thus, we have integrated our algorithms and software as a modelling tool ASTRO, which enables an investigator to generate morphological and functional astroglial features at various scales.

Finally, as a proof of concept, we explore our test-case astrocyte models (that are partly constrained by empirical data) to reveal some important aspects of astroglial physiology that are inaccessible in experiments. We therefore evaluate key electrodynamic features of the astroglial membrane, basic aspects of intracellular $K^+$ dynamics, the range of intracellular $Ca^{2+}$ buffering capacity, and how the classical molecular machinery of $Ca^2$

$^+$ 'puffs' and 'sparks' could explain some $Ca^{2+}$ imaging observations in astrocytes. Our findings suggest that ASTRO could be a valuable tool for physiological hypothesis testing and causal interpretation of experimental observations pertinent to astroglia.

## Results

**Stem tree reconstruction of live astroglia.** The gross morphology of hippocampal area CA1 astrocytes points to the cell tree radius of 30–50 μm, somatic diameter of 7–15 μm, and 4–9 primary processes[9,23–25]. To elucidate this structure further, we used acute hippocampal slices, loaded individual astroglia in whole cell with the morphological tracer Alexa Fluor 594 (Methods), and imaged the cell expanse using two-photon excitation (2PE; Fig. 1a, b). This procedure has been shown to faithfully reveal fine astroglial morphology[24].

Our modelling strategy was to start with the principal branch structure ('stem tree') which could be resolved in optical images (branch diameter above the diffraction limit, 0.3–0.5 μm; Fig. 1a, b) and then 3D-reconstructed using computer tools previously validated in neuronal studies. Focusing on one cell, we imaged it in a z-stack (Fig. 1a, b), corrected brightness of individual recorded sections for the depth-dependent signal drop, and 3D-reconstructed identifiable cell branches semi-automatically using Simple Neurite Tracer (Fiji-ImageJ, NIH; Fig. 1c; Methods). To store the recorded structure in a NEURON format (Fig. 1d) we used Vaa3D (Allen Institute). Alternatively, the entire 3D-reconstruction procedure could be carried out using commercially available Neurolucida (MBI).

Our complementary approach was to build the 'typical' stem tree representing the astrocyte pool under study. First, we used a library neurogliaform cell (P32-DEV136, [http://neuromorpho.org/neuron_info.jsp?neuron_name=P32-DEV136]) as a stem tree skeleton (Fig. 1e, diagram). Second, the numbers and diameters of its branches were adjusted to match measurements from an experimental sample of CA1 astroglia (13 cells, 98 branches; Fig. 1e).

**The endfoot.** The endfoot surrounding blood vessels is a key feature of most astrocytes. Because its morphology varies enormously, it would seem appropriate to use experimental 3D reconstructions (as in Fig. 1a–d) to incorporate it into the cell architecture. ASTRO provides a separate NEURON menu for constructing the endfoot and connecting it to the main arbour (Supplementary Note 1, ASTRO User Guide, pp 20–21). All nanoscopic process structures and biophysical mechanisms available in the present model builder (sections below) could be incorporated into the endfoot. However, as it represents a highly specialized cell compartment, it will require a separate study to develop its biophysical machinery in accord with experimental observations. Simulation tests in the present study will focus therefore on the 'main' astrocyte arbour (parts of which may include processes that surround small blood vessels).

**Experimental measurements of nanoscopic astroglial processes.** The next model-building step was stochastic generation of nanoscopic processes. The bulk of astroglial morphology comprises irregularly shaped ultrathin branchlets that appear as a blur in optical images (Fig. 1a, b). To quantify such structures, we used correlational 3D EM[26]. Individual astrocytes were filled with biocytine whole-cell, and after DAB conversion were traced and reconstructed using serial sections (Methods; Supplementary Fig. 1). It has recently transpired that chemical fixation via heart perfusion might shrink tissue by 30–35% also causing aberrations in astroglial morphology[27]. To minimise such effects, we rapidly fixed thin acute slices by submersion, which causes only ~5%

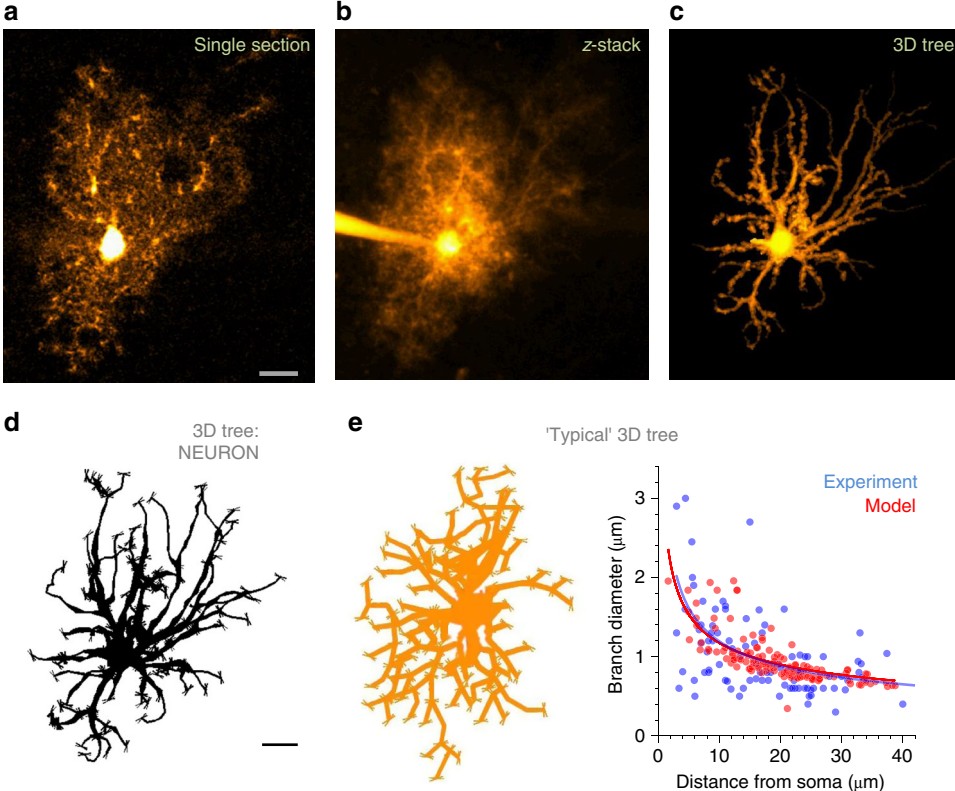

**Fig. 1** Reconstructing astroglial stem tree in silico. **a** A characteristic image of CA1 astroglia, whole-cell load with Alexa Fluor 594 ($\lambda_x^{2p} = 800$ nm), single optical section (stratum radiatum, depth of ~100 µm). Scale bar, 10 µm (applies to **a**–**c**). **b** Cell as in panel **a** shown as a full z-stack projection. **c** Stem tree of astroglia shown in **a** and **b**, separated and reconstructed in 3D using NeuroTrace (Fiji ImageJ, NIH); 2D view of a 3D image. **d** Astrocyte stem tree shown in panel **c** quantified, loaded and displayed in NEURON format using Vaa3D (Allen Institute); thin 'buds' indicate initial seeds for 'planting' nanoscopic protrusions at a certain longitudinal density; 2D view. Scale bar, 10 µm (applies to **d** and **e**). **e** Diagram, 'typical' astrocyte stem tree built by modifying a library neurogliaform cell (2D view); plot, matching the branch diameters in the model (red) and in recorded astroglia (blue; $n = 13$ cells including 98 dendrites); solid lines, the best-fit dependence (power low, $y = a \cdot x^b$) for the corresponding data scatters

linear tissue shrinkage[28] preserving the extracellular volume fraction of ~0.12[29], close to ~0.15 under cryofixation[27].

Reconstructing the entire astrocyte with 3D EM is difficult and may not necessarily represent the 'typical' cell. We instead focused on small fragments sampled from multiple CA1 astrocytes[26] (Fig. 2a), aiming to extract key statistical features of their nanoscopic processes. Specific ASTRO routines were developed to sample and store branchlets from 3D-reconstructed cell fragments (Fig. 2a, b; Supplementary Note 1, ASTRO User Guide p. 12). Each sampled process comprised varied-length stacks of 60 nm thick sections, with individual sections being represented by 3D point co-ordinates scattered on their 'polygonal' perimeters (Fig. 2c; Supplementary Note 1, ASTRO User Guide pp. 12–13). Thus, a representative sample of astrocyte nanoscopic processes was obtained.

**NEURON-compatible transformation of nanoscopic processes.** NEURON-built cell models use cylindrical compartments that follow the shape of neuronal dendrites or axons. Because astroglial processes have irregular shapes (Fig. 2a–c), we carried out a separate investigation to establish how their geometry and biophysical properties could be recapitulated using cylindrical compartments.

We therefore transformed 'polygonal' z-stacks representing 3D-reconstructed processes, into z-stacks of cylindrical slabs (Fig. 2d). Here, the adjacent polygonal slabs, with cross-section areas $S_i$ and $S_{i+1}$ and an intersection area of $T_i$ (Fig. 2d, left and middle), were represented by two 'main' cylinder slabs, with base

areas $S_i$ and $S_{i+1}$ (termed 'leaves'), plus a 'transitional slab' ('stalk'), with base area $T_i$ (Fig. 2d, right). This transformation largely preserved the diffusion bottleneck and the surface–volume relationships of the original shape. By applying this rule, we transformed all stored 3D-processes into NEURON-compatible shapes (Fig. 2e).

Next, we employed Monte Carlo simulations to systematically compare original and NEURON-compatible shapes with respect to the two key biophysical traits, diffusion transfer rate and dynamic electrical impedance. This involved 'injecting' 3000 Brownian particles into one end and monitoring them at the other end of the shape, with or without an electric field applied (Fig. 2f; example in Supplementary Movie 1). The algorithms involved were tested and validated by us against experimental data previously[30,31]. In most instances, there was a remarkable similarity between the two shape types in their properties (Fig. 2g); in the remaining cases, a minor adjustment of the cylindrical compartment diameters achieved a similar match.

Importantly, the biophysical match between the original and the cylinder-based shapes held when cylindrical compartments were shuffled randomly (Supplementary Fig. 3). Thus, the frequency distribution of cylinder diameters was sufficient to obtain nanoscopic structures biophysically compatible with real cell processes. The transformation procedure described above has been integrated in ASTRO (Supplementary Note 1, ASTRO User Guide, pp. 11–18). The next step for ASTRO was therefore to populate the cell stem tree (Fig. 1d, e) with nanoscopic processes using their experimental statistics and further empirical constrains described below.

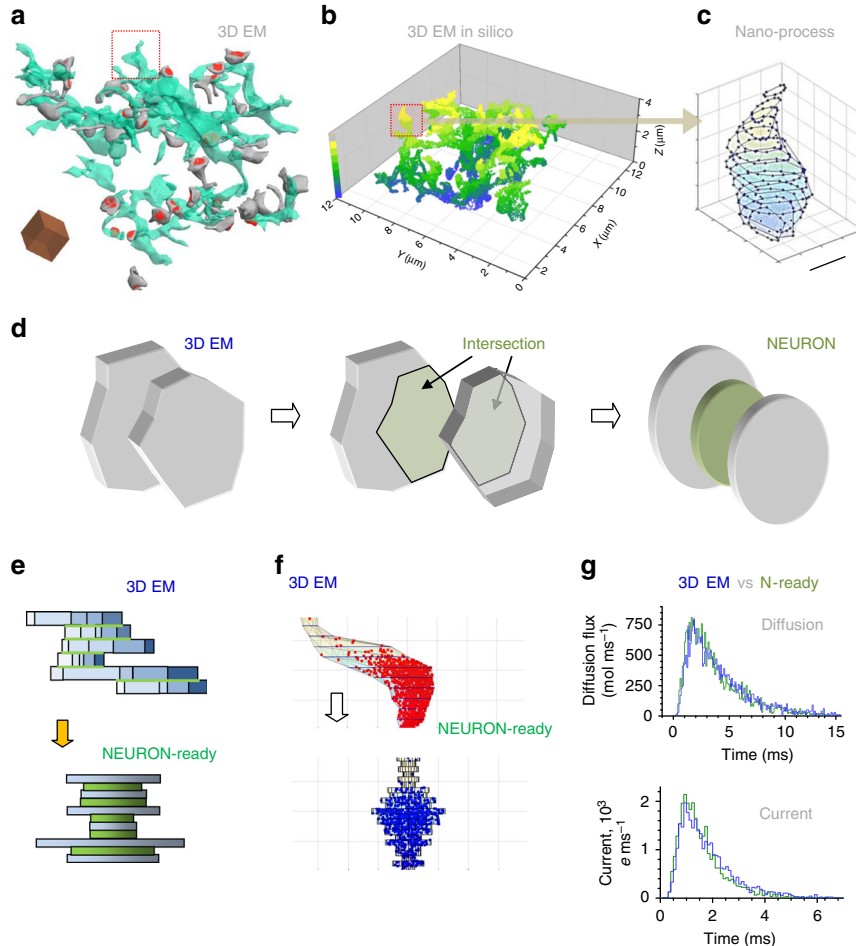

**Fig. 2** Nanoscopic astroglial protrusions: from 3D EM to in silico. **a** 3D EM serial-section reconstruction of an astrocytic fragment (green) and adjacent dendritic spines (grey) with postsynaptic densities (red) in area CA1; surface rendering applied[26]; dotted square, selected nano-process; Scale cube side, 1 μm. **b** Fragment in **a** shown using surface point scatter; false colour scale, z-depth as indicated; dotted square, selected nano-process, as in **a. c**, Selected process (highlighted in **a** and **b**) as a stack of polygonal sections (60 nm thick, to follow EM sectioning) delimited by surface points. Scale bar, 500 nm. **d** Transformation of the adjacent 3D EM sections (left; grey polygonal slabs with base areas $S_i$ and $S_{i+1}$) with intersection area $T_i$ (middle; green polygons) into NEURON-compatible two main (grey, 'leaves') and one transitional (green, 'stalk') cylindrical slabs with the corresponding base areas (right). **e** Transformation of 3D EM reconstructed processes (top) into NEURON-compatible cylinder section stacks (bottom). Individual sections (blue, top) are transformed into 'main' cylinders (blue, bottom), and green segments (top) depict adjacent surfaces between sections represented by green 'transitional' cylinders (bottom), as in **d**. **f** A characteristic example of a 3D EM reconstructed astroglial process made up by its serial polygonal sections (top) and its representation by serial cylindrical compartments (bottom). Scattered dots illustrate a snapshot of the Monte Carlo simulation test (monitored live in ASTRO; Supplementary Movie 1) in which Brownian particles are injected into the bottom of the 3D structure, and their arrival time at the top is registered, to compare molecular diffusivity (no electric field) and electrodynamic properties ($2.5 \times 10^3$ V m$^{-1}$ electric field in the z-direction, one electron charge $e = 1.6 \times 10^{-19}$ C per particle applied) between the two shapes. **g** The outcome of Monte Carlo tests comparing two shapes shown in **f**, for the molecular diffusion flux (top) and ion current (bottom), measured at the top exit of the shapes (as in **f**), upon injection of the Brownian particles into the bottom entry (as in **f**); blue and green, 3D EM reconstructed and NEURON-compatible shapes, respectively. See ref. [31] for electrodiffusion simulation detail

**Tissue volume fraction and surface-to-volume ratios**. The tissue volume fraction (VF) occupied by astroglial processes in the hippocampal neuropil ranges between 5 and 10%[26,29,32,33]. The VF distribution provides a key descriptor of astroglial morphology because individual astrocytes occupy adjacent tissue domains with little overlap while their processes fill the volume in a sponge-like manner[23,34]. 2PE microscopy enables the direct monitoring of VF in live astroglia in situ because it collects emission within a thin focal layer only (Supplementary Fig. 4a). Thus, fluorescence intensity of the dye-filled astroglia scales with VF occupied by local astroglial processes (Supplementary Fig. 4b) whereas somatic cytosol imaged in the same focal plane corresponds to ~100% VF (Supplementary Fig. 4c)[26,35]. Thus, the local-to-somatic emission ratio can provide direct readout of

astroglial VF (Fig. 3a). We thus obtained the VF distribution within individual CA1 astrocytes (Fig. 3b) and used it to constrain stochastic generation of nanoscopic processes in the model.

The latter is achieved by adjusting two parameters in the ASTRO-NEURON menu: the average size of nanoscopic processes (number of leaves per process, <100; Supplementary Note 1, ASTRO User Guide, Fig. 16, p. 19), and their 'seed density' (normally between 1 and 3 per branch; ASTRO SeedNumber parameter; Supplementary Note 1, ASTRO User Guide, p. 20). Another important feature of cell processes is their surface-to-volume ratio (SVR): it determines how transmembrane fluxes are converted into intracellular concentration dynamics. Stereological analyses of hippocampal astroglial processes using 3D EM estimates SVRs in the 15–25 μm$^{-1}$

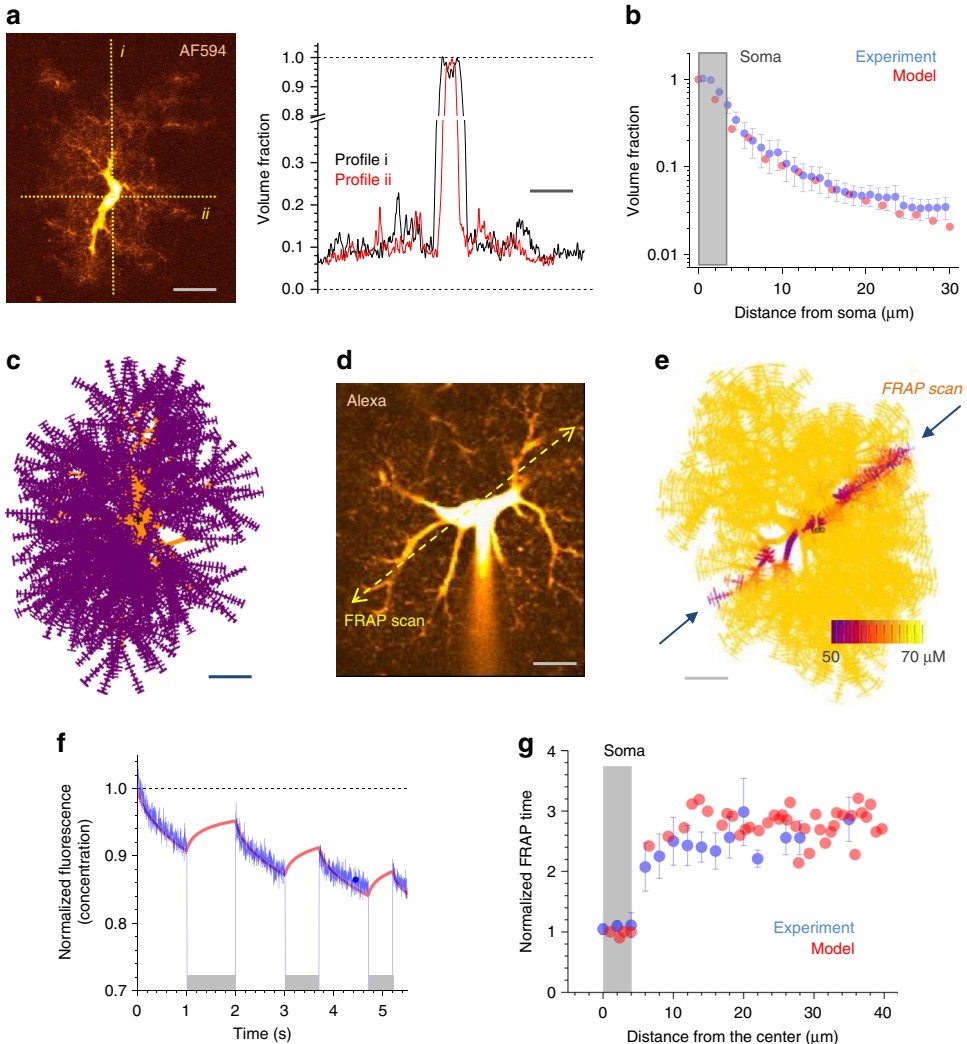

**Fig. 3** NEURON-based astrocyte model: determining volumetric quantities. **a** Image panel, a characteristic astrocyte in area CA1 (Alexa Fluor 594, $\lambda_x^{2p} = 800$ nm) seen in a single 2PE optical section (~1 µm thick) crossing the cell soma; dotted lines, sampling fluorescence intensity profiles reporting the astrocyte tissue volume fraction (VF); scale bar, 15 µm. Graph, VF profiles (fluorescence local/soma ratio) obtained along the dotted lines *i* and *ii* in the image, as indicated; scale bar, 10 µm. **b** Matching modelled (red) and experimental (blue; mean ± SEM; *n* = 13 astrocytes) VF values (ordinate, dimensionless) sampled at different distances from the soma (abscissa). **c** A complete NEURON-generated astrocyte model (z-projection), with main branches depicted in orange (partly obscured by smaller processes) and nanoscopic protrusions (schematic depiction) in purple. Note that tortuous processes of real-life astroglia are represented here by biophysically equivalent 'straightened' processes. Scale bar, 10 µm. **d** An example of astroglia as in **a**; dotted line, line-scan position to measure internal diffusion connectivity (using Alexa Fluor 594 photobleaching); patch pipette tip is seen. Scale bar, 10 µm. **e** A snapshot of a photobleaching experiment in silico showing the intracellular Alexa concentration dynamics in a modelled astrocyte; arrows, photobleaching line positioning; false colour scale, intracellular Alexa concentration, as indicated (Supplementary Movie 2). **f** Matching the modelled (red) and the experimental (blue) time course of intracellular Alexa Flour fluorescence during a photobleaching experiment as shown in **d** and **e**, one-cell example (CA1 area, stratum radiatum astrocyte). Grey segments indicate laser shutter-on when fluorescence recovery occurs (red). **g** Statistical summary of photobleaching experiments (*n* = 10 astrocytes) and related simulations, as depicted in **d** and **e**, comparing experimental (blue) and simulated (red) data

range[9,29,33,36]. The astrocyte fragments sampled in our experiments (Fig. 2a–c) had SVRs within this range, which was faithfully reproduced by the cylinder-based shapes (Fig. 2f bottom, Supplementary Fig. 3b). ASTRO could further fine-tune the average SVR, by adjusting the distribution of 'leaves' and 'stalks' in stochastically simulated nanoscopic processes (Supplementary Note 1, ASTRO User Guide, pp. 21–23). With these steps completed, we arrived at realistic, NEURON-compatible CA1 astrocyte geometry comprising 35,000–45,000 individual compartments (Fig. 3c; Supplementary Table 1).

**Astroglial internal connectivity in experiment versus model.** Our final check was to see if the model has intracellular diffusion

connectivity similar to that in live astroglia. To test that, we used the fluorescence-recovery-after-photobleaching (FRAP) approach: the fluorescence recovery rate in a laser-bleached cell region reflects diffusion speed of the fluorescent molecules. We therefore sought to gauge diffusion connectivity among astroglial compartments by bleaching dye molecules within a thin cylindrical volume (laser line-scan) across the astrocyte arbour (Fig. 3d; Methods)[37].

First, we confirmed that fluorescence fully recovered within 60 s of the FRAP cycle (Supplementary Fig. 5a,b), with the similar kinetics in subsequent FRAP cycles (Supplementary Fig. 5c), thus pointing to the FRAP stability in our settings. Next we simulated a FRAP protocol in the model (Fig. 3e; Supplementary Movie 2)

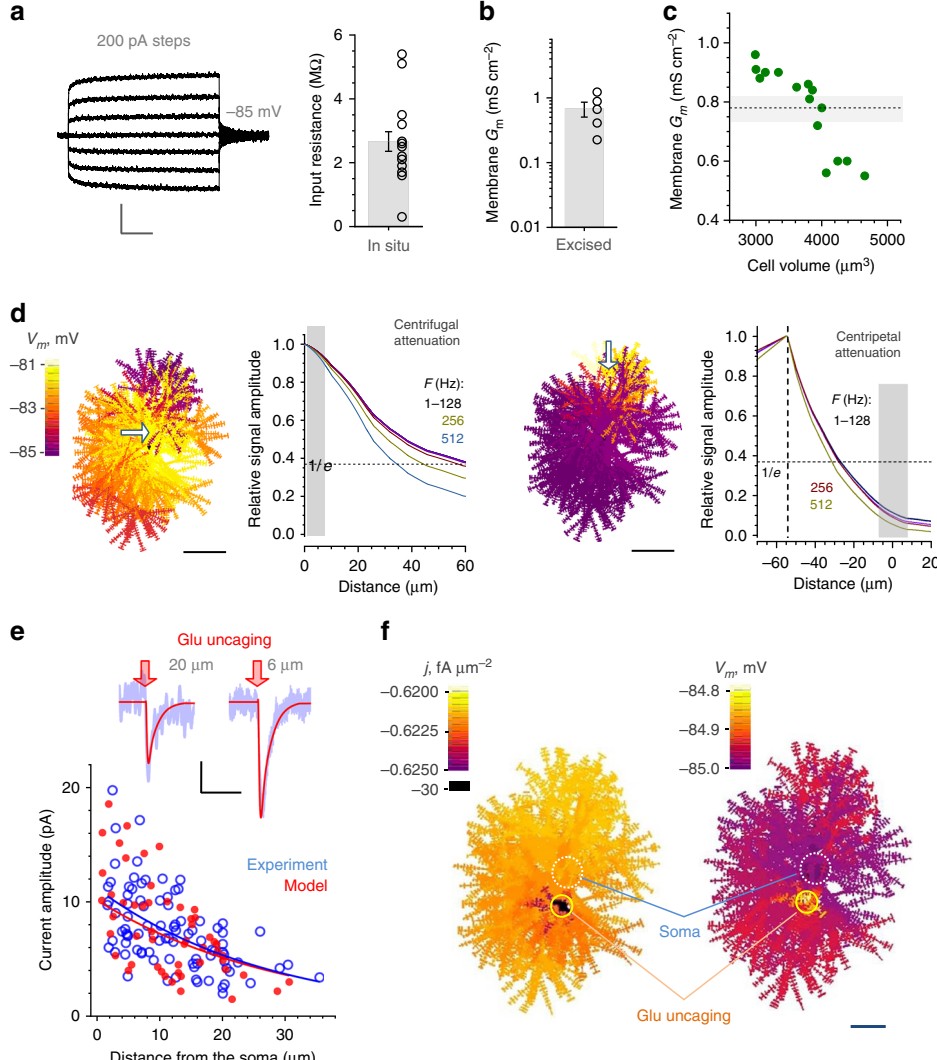

**Fig. 4** Electrogenic properties of protoplasmic astroglia. **a** Traces, a characteristic current-voltage recording of CA1 astroglia; graph, input resistance (bar, mean ± SEM; dots, individual cell data; $n = 15$). Scale bars (v, h): 1 mV, 100 ms. **b** Specific membrane conductance $G_m$ measured in excised whole-cell (outside-out) patches of CA1 astrocytes (bar, mean ± SEM; dots, individual cell data; $n = 5$). **c** Dots, $G_m$ values obtained from Ohm's law $G_m = (S_{mem}R_i)^{-1}$ in stochastically generating astrocyte models within the empirical range of cell volumes (abscissa) and input resistance matching data shown in **a**; dotted line and grey shade, mean ± SEM for the sample shown; note that NEURON-model astrocyte surface area accounts for both sides and bases of individual cylindrical compartments (Methods). **d** Membrane space constant estimated using a full astrocyte model for centrifugal (left panels) and centripetal (right panels) voltage signal propagation. Cell shape diagrams: $V_m$ landscape snapshots generated by local application (shown by arrow) of a sine voltage signal (amplitude + 5 mV). Graphs: signal amplitude attenuation at various signal frequencies, as indicated, for centrifugal and centripetal cases, as indicated. Scale bars, 20 μm. **e**, Traces, example of whole-cell recordings (blue) in response to spot-uncaging of glutamate ($\lambda_u^{2p} = 720$ nm, 20 ms duration), at two distances from the astrocyte soma, as indicated; red lines, simulated whole-cell current in the corresponding model arrangement (~5 μm wide glutamate application; GLT-1 kinetics;[43,73] GLT-1 surface density $10^4$ μm$^{-2}$ as estimated earlier [44]). Plot, a summary of glutamate uncaging experiments (blue open dots, $n = 8$ cells/90 uncaging spots) and uncaging tests simulated in the model (red solid dots, $n = 39$). Scale bars (v, h): 2 pA, 150 ms. **f** Model snapshot 5 ms post glutamate spot-uncaging depicting the cell membrane current density ($j$, left) and voltage ($V_m$; right) landscape (example in Supplementary Movie 3); false colour scale. Scale bar, 10 μm

comparing the modelled outcome with experimental observations. By adjusting one free model parameter (photobleaching rate), we were able to match experimental and simulated data in individual cells (Fig. 3f) and across the sample ($n = 10$; Fig. 3g). Thus, the modelled cell faithfully represented inner connectivity of CA1 astrocytes.

**Passive electrical properties of astrocytes.** In whole-cell configuration, input resistance $R_i$ of CA1 astrocytes in our sample ($2.66 \pm 0.31$ MΩ, mean ± SEM, $n = 15$; Fig. 4a) was consistent with the previous reports[20,21]. To assess specific membrane conductance $G_m$, we measured resistance of outside-out patches and estimated the patch area using the classical voltage-step method[38] (Methods), which gave $G_m = 0.69 \pm 0.18$ mS cm$^{-2}$ (mean ± SEM; Fig. 4b). Next we asked if the model could actually predict experimental $G_m$ (Fig. 4b) when two other model parameters, $R_i$ and total membrane area $S_{mem}$, were empirically constrained. We have therefore produced a small representative sample of same-type astrocyte models, by repeating stats-constrained stochastic generation of nanoscopic processes on the same stem tree (Fig. 4c). In the sample, the cell volume varied well within the range of 3000–4800 μm$^3$ characteristic of CA1 astroglia[23,29]. In each sampled cell model, $G_m$

was a free parameter which was adjusted until model $R_i$ matched its empirical value (Fig. 4a). This test produced the average $G_m$ value of $0.78 \pm 0.04$ mS cm$^{-2}$ (Fig. 4c), which was indistinguishable from the experimental $G_m$ value (Fig. 4b) or its earlier measurements[39]. This results indicates that stochastic generation of nanoscopic processes produces realistic membrane properties of astrocytes.

We next simulated a voltage-clamp experiment, showing that large currents in these leaky cells induce only small somatic depolarisation while propagating with a space constant of 30–60 μm depending on the signal frequency (Fig. 4d, left). Voltage signals generated at the cell periphery have, due to the 'cable-end' proximity, a shorter constant of 25–30 μm (Fig. 4d, right) suggesting that similar signals at different astrocyte loci could have different membrane effects.

Astrocyte membranes are enriched in potassium channels, in particular K$_{ir}$4.1 type[39,40]. Their typical unit conductance is either compatible with or lower than the membrane current leak due to the other conductances (channels, exchanges, gap junctions)[17,41] which were represented, for the sake of simplicity, by a non-specific channel current maintaining resting $V_m$ near −83…−85 mV. Thus, K$_{ir}$4.1 should have little effect on the voltage spread profile, even though these channels largely control the cell resting membrane potential[42]. Indeed, when we added evenly distributed K$_{ir}$4.1, their effect on voltage was detected only when their overall conductance exceeded its expected physiological range (Supplementary Fig. 6a). In physiological circumstances, however, this scenario could be affected by changes in extracellular K$^+$, and by the poorly understood contribution of other ion channels and exchangers (see below).

**Voltage–current landscape generated by glutamate uptake**. To understand how the current generated by glial glutamate transporters (GLT-1) affects membrane potential across the cell, we first recorded from a CA1 astrocyte the response to two-photon spot-uncaging of glutamate (20 ms) at variable distances from the soma (Fig. 4e, traces). Second, we replicated this test in the model, by implementing the GLT-1 kinetics[43], scattering the transporters uniformly on the cell surface (at ~10$^4$ μm$^{-2}$)[44], and applying extracellular glutamate within small spherical areas (radius ~3 μm, duration ~20 ms) at quasi-random distances from the soma. An excellent match between the modelled somatic current and whole-cell recording data could be obtained (Fig. 4e, plot) by adjusting one free model parameter, the amount (peak concentration) of released glutamate. The model unveiled the dynamic landscape of astrocyte membrane voltage, which varied only within ~0.2 mV across the entire cell (Fig. 4f; Supplementary Movie 3).

These simulations illustrate quantitatively that a local current hotspot in the electrically leaky astroglial membrane stays localized, with little effect on the voltage landscape (Fig. 4f). Again, this scenario was only weakly affected by adding K$_{ir}$4.1 (Supplementary Fig. 6b-d) although in real cells the effect could be more complex. In any case, these tests suggest that GLT-1 currents on their own cannot significantly depolarise astrocyte membrane away from the region of active transport.

**Potassium uptake and redistribution inside astroglia**. Rapid uptake and intracellular redistribution of potassium are essential functions of brain astroglia. Several models have dealt with this mechanism on a cell or tissue level[17,45,46], and here we aimed to understand its complex dynamics inside astroglia. As a proof of concept, we simulated a scenario of intense local activity in which extracellular K$^+$ concentration [K$^+$]$_{out}$ was elevated for two seconds, from 3 to 10 mM, inside a 20 μm spherical tissue area

(Fig. 5a). The model was populated with K$_{ir}$4.1 channels[17] with unit conductance of 0.1 mS cm$^{-2}$. The [K$^+$]$_{out}$ elevation activated K$_{ir}$4.1 homogeneously inside the 20 μm area, prompting K$^+$ entry (peak current density ~0.01 mA cm$^{-2}$). The ensuing local increase in intracellular K$^+$ concentration [K$^+$]$_{in}$ (from 110 to ~113 mM) dissipated over several seconds after [K$^+$]$_{out}$ returned to 3 mM (Fig. 4b). The period of elevated [K$^+$]$_{out}$ also featured very slight depolarisation generated by K$_{ir}$4.1 (Fig. 5c), not dissimilar to that arising from glutamate uptake (Fig. 4f). These data suggest that extracellular K$^+$ buffering and its intracellular redistribution could be controlled by local K$^+$ efflux, in particular through K$_{ir}$4.1. However, further experimental constrains are required to understand the possible contribution of other membrane mechanisms.

To dissect theoretically a possible role of active K$^+$ extrusion mechanisms, such as pumps and ion exchangers, we carried out similar tests but with the active removal of intracellular K$^+$ by a first-order pump (which may also reflect gap-junction escape) and no contribution from K$_{ir}$4.1 channels (Supplementary Fig. 7). The dynamic spatial landscape of [K$^+$]$_{in}$ captured by the model confirmed that local K$^+$ efflux could efficiently limit the spatial spread of [K$^+$]$_{in}$ elevations (Supplementary Fig. 7).

**Gap junctions and hemichannels**. Neighbouring astroglia are connected via gap junctions (made up by adjoined connexin proteins), enabling current leak and diffusional flow of molecules across the astroglial syncytium[47]. Astroglia are also enriched in connexin hemichannels permitting molecular transfer to and from the extracellular medium[48]. In our tests, blocking these channels with carbenoxolon (CBX, 50 μM) increased $R_i$ by ~30% (Supplementary Fig. 8), consistent with previous reports[49].

We have incorporated basic gap junction options in the ASTRO menu, both as electric conductance and as a diffusion channel (Methods; Supplementary Note 1, ASTRO User Guide, pp. 18, 26–28). For the sake of clarity, however, simulations here consider gap junctions as a constituent contributor to membrane conductance. Dissecting their precise roles will require further experimental detail pertinent to their biophysics and their intercellular distribution.

**Probing the impact of calcium buffering on calcium waves**. Astroglial Ca$^{2+}$ waves are thought to rely on Ca$^{2+}$ stores, channels, and pumps involving endoplasmic reticulum and mitochondria. The underlying molecular machinery appears to engage Ca$^{2+}$-dependent Ca$^{2+}$ release controlled by inositol 1,4,5-trisphosphate (IP$_3$) and possibly ryanodine receptor-channels displaying a highly non-linear (sometimes bell-shaped) dependence between channel activity and Ca$^{2+}$ concentration[50]. Recently, local Ca$^{2+}$ transients (but not global waves) have been documented in astroglia lacking IP$_3$ receptors[51,52]. In any such cases, intracellular Ca$^{2+}$ signal propagation must be tightly controlled by the local Ca$^{2+}$ buffering capacity[53], the feature comprehensively explored in nerve cells.

The NEURON environment by default incorporates biophysical mechanisms of Ca$^{2+}$ dynamics including diffusion, buffering, and IP$_3$ action (Methods), to which we have added two further IP$_3$-dependent mechanisms described in the literature[14,54]. In the default configuration, four parameters control intracellular Ca$^{2+}$ waves: resting IP$_3$ concentration $C_{IP3}$, resting Ca$^{2+}$ concentration [Ca$^{2+}$]$_{rest}$, endogenous Ca$^{2+}$ buffer concentration [B] and its affinity (dissociation constant) $K_D$. Most studies constrain $C_{IP3}$ within 0.5–3 μM[55] whereas earlier we were able to measure [Ca$^{2+}$]$_{rest}$ using time-resolved fluorescence microscopy (range 50–100 nM)[56]. In contrast, [B] and $K_D$ can vary widely across cell types and thus remain unknown.

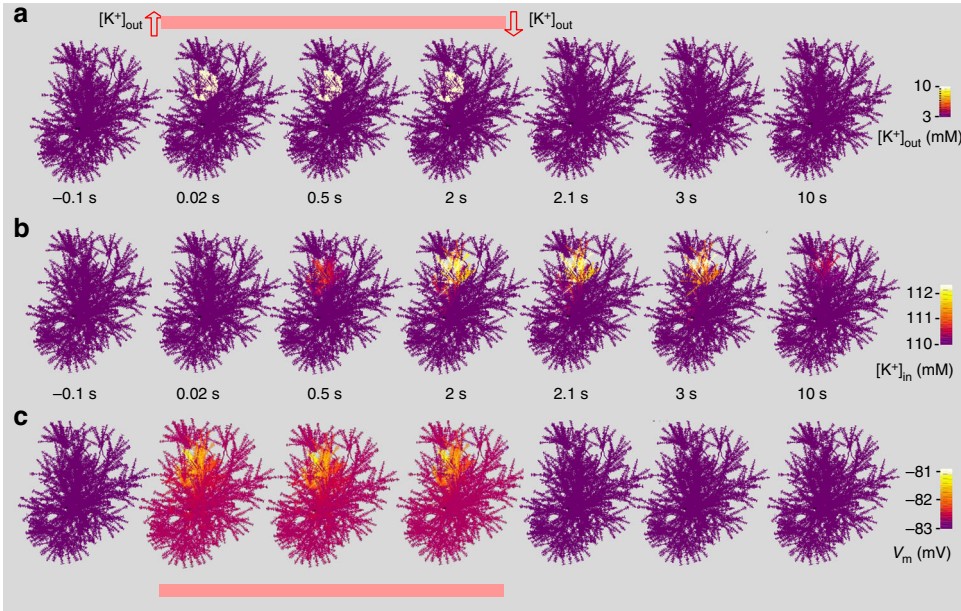

**Fig. 5** Cellular dynamics triggered by extracellular potassium rise. **a** Cell shape diagrams, time series snapshots of the cell shape (3D-reconstruction reconstruction shown in Fig. 1a–d) illustrating a spherical 20 μm wide area within which extracellular $[K^+]_{out}$ was elevated from baseline 3 to 10 mM, for 2 s (onset at $t = 0$), as indicated; $K_{ir}4.1$ channels were evenly distributed with unit conductance of 0.4 mS cm$^{-2}$ (no other leak conductance) generating peak current density (in the region with $[K^+]_{out} = 10$ mM) of 0.01 mA cm$^{-2}$. The $K_{ir}4.1$ kinetics were incorporated in NEURON, in accord with ref. [17], as $I_{Kir} = G^*_{K_0}(V_A - V_{KA} - V_{A1})\sqrt{[K^+]_{out}}\left(1 + \exp\left(\frac{V_A - V_{KA} - V_{A2}}{V_{A3}}\right)\right)^{-1} + I_{LA}$ where $G^*_{K_0}$ is the effective conductance factor, $V_{KA}$ is the Nernst astrocyte K$^+$ potential, $V_A$ astrocyte membrane potential, $K_O$ is $[K^+]_{out}$, $V_{A1}$ an equilibrium parameter (sets $I_{Kir}$ to 0 at $-80$ mV), $V_{A2}$ and $V_{A3}$ are constants calibrated by the $I$–$V$ curve, and $I_{LA}$ residual leak current. **b** Cell shape diagrams, snapshots illustrating the spatiotemporal dynamics of internal $[K^+]_{in}$ in the test shown in **a**; false colour scale, as indicated. **c** Snapshots illustrating the spatiotemporal dynamics of the membrane voltage in the test shown in **a**; false colour scale, as indicated

As expected, the model could generate Ca$^{2+}$ waves over a wide (physiologically plausible) range of the above parameters. To dissect the basic effect of Ca$^{2+}$ buffering on Ca$^{2+}$ wave propagation, we compared wave dynamics with and without a small amount (10 μM) of mobile Ca$^{2+}$ buffer added (Fig. 6a; Supplementary Movie 4). Adding the buffer appeared to significantly reduce the wave speed and amplitude (Fig. 6a), suggesting that increased Ca$^{2+}$ buffering may actually prolong periods of elevated [Ca$^{2+}$]. However, accurate interpretation of such results requires further empirical constrains.

**Assessing calcium-buffering capacity in vivo.** Because there is an ongoing debate on whether astroglial Ca$^{2+}$ waves seen in acute slices are fully physiological[57], we sought to document such waves in live animals. However, imaging hippocampal astrocytes in vivo involves mechanical invasion the impact of which on astroglial function is not fully understood. We therefore imaged somatosensory cortex astroglia (accessible with intact brain surface), which appears remarkably similar in their basic morphological features and territorial volumes to hippocampal astrocytes[58].

We asked whether [B] and $K_D$ could be assessed by matching the empirical Ca$^{2+}$ wave dynamics to the modelled outcome. Spontaneous activity of astrocytes in vivo was recorded using a virus-transduced Ca$^{2+}$ indicator expressed under a GFAP promoter: to minimize filtering effects of free-diffusing Ca$^{2+}$ indicators, we used the plasma-membrane-tethered GCaMP6f (~50 ms fluorescence response time)[59]. The animals were anaesthetized, to limit bursts of sensory input-evoked prominent Ca$^{2+}$ rises that could be mistaken for self-propagating Ca$^{2+}$ waves. Gross morphology of astrocytes was monitored in the red channel using bolus-loading of sulforhodamine 101 (Fig. 6b).

In baseline conditions, spontaneous Ca$^{2+}$ waves engulfing individual cells (spread over 10–20 μm) appeared on average ~5.4 times a minute over a 160 μm x160 μm ROI (Fig. 6c, top; Supplementary Movie 5), in good correspondence with previous observations[60]. The time first derivative of the fluorescence transients revealed their wave-front dynamics (Fig. 6c, bottom) helping to identify centrifugally spreading events and thus to distinguish single-cell regenerative waves from synchronous signals evoked by external influences.

We found that such waves propagated with an average radial speed of $3.94 \pm 0.16$ μm s$^{-1}$ (mean ± SEM, $n = 54$ events). Intriguingly, this speed appears significantly lower than that of the stimulus-induced astroglial Ca$^{2+}$ waves in brain slices or in culture (15–25 μm s$^{-1}$, reviewed in ref. [61]). One possible explanation is that an exogenous stimulus in situ, such as agonist application, is likely to trigger a synchronous receptor response over the entire cell expanse. Similarly, in awake animals, we detected single-cell Ca$^{2+}$ waves resembling those under anaesthesia, albeit at lower frequencies and magnitudes, in between prominent, region-wide Ca$^{2+}$ elevations (Supplementary Movie 6). Possible contamination with such events made the anaesthetized animals a preferred choice in assessing Ca$^{2+}$ buffering properties of astroglia.

In the astrocyte model, experimentally observed Ca$^{2+}$ waves could be readily reproduced using an instantaneous local Ca$^{2+}$ rise (5 μM for 0.1 ms) near the soma (Fig. 6d). Although the model could generate waves over a wide range of Ca$^{2+}$ buffering parameters, their experimental speed required only certain combinations of [B] and $K_D$ (Fig. 6e) reflected in an almost perfectly linear relationship (in μM): $[B] = 170(1 + K_D)$. This

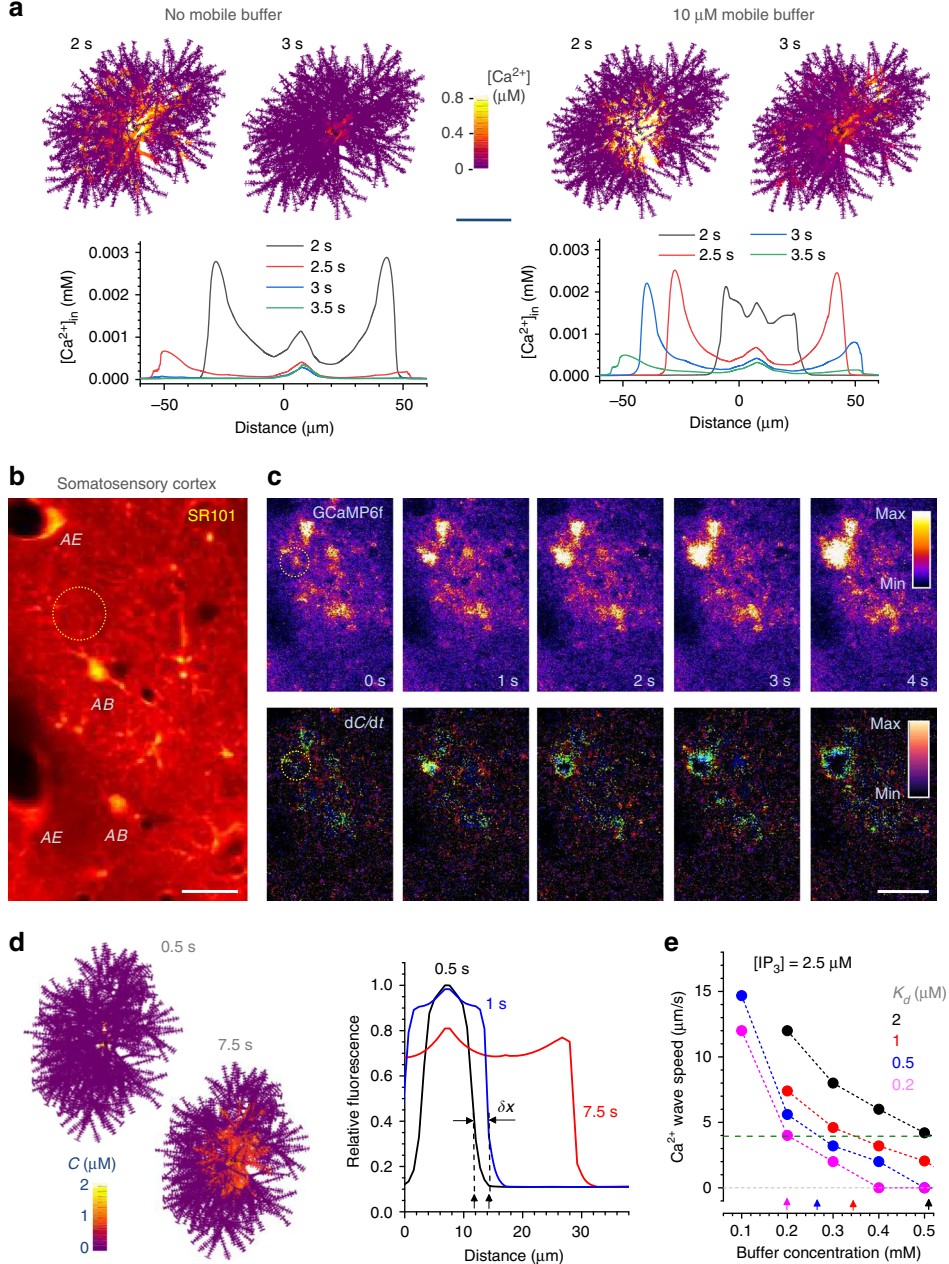

**Fig. 6** Ca²⁺ waves and Ca²⁺ buffering capacity of astrocytes. **a** Cell diagrams, [Ca²⁺] landscape snapshots (some branches obstruct full 3D view) at time points after wave generation, with and without Ca²⁺ buffer, as indicated; graphs, [Ca²⁺] dynamics snapshots (zero Distance, soma centre), as indicated. Model parameters: Ca²⁺ diffusion coefficient, 0.3 μm² ms⁻¹; immobile/endogenous Ca²⁺ buffer concentration, 200 μM ($K_f = 1000$ mM⁻¹ ms⁻¹; $K_D = 20$ ms⁻¹); mobile Ca²⁺ buffer concentration, 10 μM ($K_f = 600$ mM⁻¹ ms⁻¹; $K_D = 0.5$ ms⁻¹; $D = 0.05$ μm² ms⁻¹); Ca²⁺ pump activation threshold, 50 nM; Ca²⁺ pump flux density, 20 μM ms⁻¹; basal IP₃ concentration, 0.8 μM; IP₃ concentration upon release, 5 μM (onset, 1 s; further detail in Supplementary Note 1, ASTRO User Guide, Supplementary Movie 4). Scale bar, 30 μm. **b** Rat somatosensory cortex in vivo (~100 μm deep) single 2PE optical section, bolus-loading with sulforhodamine 101 to label astroglial structures[56]; *AB* and *AE*, examples of astrocyte somata and endfoot processes, respectively. Scale bar, 15 μm. **c** Region of interest (as in **b**) in the GCaMP6f (green) channel. Top, snapshot sequence (Supplementary Movie 5; awake-animal example in Supplementary Movie 6) depicting an intracellular Ca²⁺ wave (dotted circle); bottom, same sequence shown as the time derivative (over 50 ms interval) highlighting Ca²⁺ wave front; false colour scale. Scale bar, 30 μm. **d** Cell diagrams, snapshots of Ca²⁺ wave spreading with the speed that matches experimental observations; false colour scale (C, concentration). Plot, intracellular [Ca²⁺] profile depicting the centrifugal Ca²⁺ wave propagation (seen in vivo); δx illustrates wave speed measurement (distance travelled over 0.5 s). **e** Summary: estimated combination of Ca²⁺ buffer affinity ($K_d$) and concentration that correspond to the observed Ca²⁺ wave speed; [IP₃], assumed intracellular concentration of IP₃[15,55,74]; horizontal dotted line, average experimental speed of astroglial Ca²⁺ waves in vivo (as in **c**; $n = 54$ events in ~20 cells)

simple formula captures the buffering properties of cortical astrocytes, also suggesting the lower-limit Ca²⁺ buffer concentration of ~170 μM (ignoring the residual effect of GCaMP6f).

**Decoding fluorescent Ca²⁺ signals recorded in astroglia.** Historically, global and slow Ca²⁺ elevations had been the key indicator of astroglial activity. Recent advances in Ca²⁺ imaging

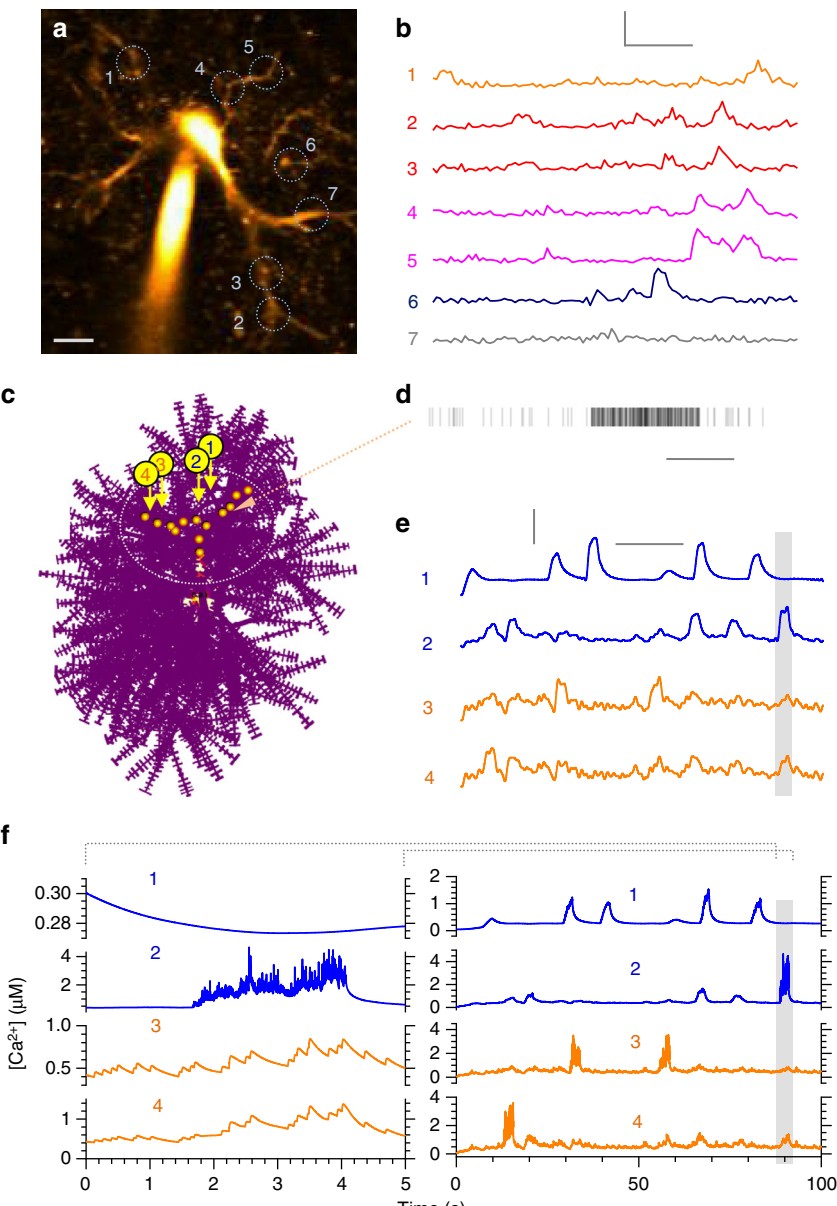

**Fig. 7** $Ca^{2+}$ dynamics decoded from fluorescence $Ca^{2+}$ imaging in situ. **a** Example, astrocyte (CA1 area, acute hippocampal slice; Fluo 4 channel, $\lambda_x^{2P} = 800$ nm) held in whole cell, with regions of interest for $Ca^{2+}$ monitoring (circles, ROIs 1–7; Supplementary Movie 7). Scale bar, 5 µm. **b** Time course of $Ca^{2+}$ sensitive fluorescence (Fluo-4 channel) recorded in ROIs 1–7 as in **a**, over 100 s; same colours correspond to ROIs on the same branch (ROIs 2–3 and 4–5). Scale bars (v, h): 200% $\Delta F/F$, 20 s. **c** An astrocyte model with localized $Ca^{2+}$-puff sources (orange dots) and four recording points (arrows, 1–4); dotted oval, region for analyses: cell area outside has a negligible effect of the $Ca^{2+}$ sources as shown (Supplementary Movie 8). The model is 'filled' with free-diffusing Fluo-4 (150 µM; $k_{on} = 600$ mM$^{-1}$ ms$^{-1}$, $k_{off} = 21$ ms$^{-1}$) and the endogenous buffer as estimated (Fig. 6d; 200 µM, $K_D = 0.2$ µM; other combinations produced similar results). **d** Example of channel-like local $Ca^{2+}$ entry activity generated by a single localized $Ca^{2+}$ source, in accord with the known biophysical properties of cellular $Ca^{2+}$ sparks and hotspots. Scale bar, 1 s. **e** Time course of simulated Fluo-4 fluorescence (150 µM 'added') in ROIs 1–4 shown in **c**: it has statistical properties similar to those recorded in situ (**b**); shaded area, time window for higher temporal resolution (see **f**); same line colours correspond to ROIs on the same cell branch. Scale bars (v, h): 100% $\Delta F/F$, 20 s. **f**, Right, simulated intracellular $[Ca^{2+}]$ dynamics underlying Fluo-4 fluorescence shown in **e**. Left, trace fragments on the expanded time scale (shaded area in **e**), as indicated; the fragments correspond to the period of relatively high $[Ca^{2+}]$

revealed faster and more local $Ca^{2+}$ signals prevalent in small astrocyte processes (reviewed in refs. [4,62–64]). Any such observations rely however on high-affinity $Ca^{2+}$ indicators, which provide only a crude reference to the underlying $Ca^{2+}$ signals[35]. To translate recorded fluorescence into $Ca^{2+}$ dynamics, one has to model $Ca^{2+}$ entry, diffusion, and buffering (by endogenous proteins as well as $Ca^{2+}$ indicators), as demonstrated in numerous studies of nerve and muscle cells. We therefore used ASTRO

to apply a similar modelling strategy to CA1 astrocytes loaded with Fluo-4, which show robust, multi-scale spontaneous $Ca^{2+}$ activity in acute slices (Fig. 7a, b; Supplementary Movie 7).

While the precise mechanism underlying astroglial $Ca^{2+}$ signalling is still poorly understood, there has been a large body of work exploring the biophysical machinery of $Ca^{2+}$ oscillations in other cell types (reviewed in ref. [65]). The key features emerging from these studies are the occurrence of $Ca^{2+}$ micro-domains

(0.5–5 μm apart) represented by clusters of stochastically activated $Ca^{2+}$ receptor channels (including $IP_3$ or ryanodine receptors), with the agonist-dependent mean opening time of 2–20 ms, peak amplitude of ~5 pA, and an inter-opening interval varying between 0 and 50 ms[66–68]. Stochastic activation of this system produces local $Ca^{2+}$ 'sparks' (reflecting ryanodine-receptor clusters) or 'puffs' (reflecting $IP_3$-receptor clusters), typically with an event frequency of 0.1–2 Hz[69], sometimes leading to global $Ca^{2+}$ rises. The kinetic properties of other potential $Ca^{2+}$ sources in astroglia, such as mitochondria and other $Ca^{2+}$ channels, remain to be established.

In keeping with the basic paradigm, we have scattered local clusters of $Ca^{2+}$ channels ($IP_3$ receptor type) along several branches of the modelled astrocyte, 1–5 μm apart (Fig. 7c), within a ~20 μm area of interest: $Ca^{2+}$ activity outside the area was of little consequence because of rapid diffusional dissipation in the absence of global $Ca^{2+}$ events. Next, we 'filled' the model with Fluo-4 (details in Fig. 7c) and explored simulated channel openings, within and among $Ca^{2+}$ channel clusters, over the plausible range of their characteristic frequencies (Fig. 7d). Thus, $Ca^{2+}$-dependent fluorescence (represented by the concentration of Ca-bound Fluo-4) was recorded at four arbitrarily selected points in the area of interest (Fig. 7c; Supplementary Movie 8).

We found that at an average interval between channel openings of ~3 ms within clusters, and ~7 s among clusters, the simulated fluorescence signals (Fig. 7e) were similar to the experimental recordings (Fig. 7b). The intracellular $Ca^{2+}$ dynamics underlying these signals was readily revealed by the model (Fig. 7f). Constraining this dynamics further would require experimental probing involving varied $Ca^{2+}$ buffering conditions and molecular dissection of the $Ca^{2+}$ cascades involved. Nonetheless, this example indicates that the classical molecular machinery of $Ca^{2+}$ signalling long explored in other cell types could explain $Ca^{2+}$ imaging data collected in astrocytes. Our further exploration of the $IP_3$ arrangement within astrocyte branches revealed a complex relationship between inter-cluster distances, spontaneous $Ca^{2+}$ activity and its fluorescent-indicator readout (see below).

**Probing impact of variable astroglial features on function.** One advantage of cell modelling is the possibility to predict theoretically the impact of a specific cellular or environmental feature on cell's behaviour. This approach could also reveal whether certain feature combinations can make the modelled cell biophysically unstable. We therefore carried tests in which some key functional traits of modelled cells were monitored against changes in model parameters.

First, to see how strongly gross cell morphology could influence its biophysics, we compared two different modelled cells, one with the stem tree reconstructed in an experiment (Fig. 1a–d), and the 'typical CA1 astrocyte', with the stem tree adjusted to match the average features of CA1 astrocytes (Fig. 1e). The two cells thus featured different stem trees but were populated with nanoscopic processes based on the same statistical constraints (as in Figs. 2 and 3). The test revealed only subtle differences between the cells in their membrane voltage spread, input resistance, or $Ca^{2+}$ wave generation (Supplementary Fig. 9). In a similar context, simulating astrocyte swelling by ~20% (by evenly increasing the cell process width throughout) had only moderate consequences (Supplementary Fig. 10). These examples suggest a relatively narrow range of effects arising from morphological variations per se, when all other features remain unchanged.

We next asked how intracellular diffusivity would affect the molecule equilibration time across the astrocyte (e.g., during whole-cell dialysis). Simulations mimicking somatic dialysis

showed how reducing diffusivity (from 0.6 to 0.05 $\mu m^2\,ms^{-1}$, reflecting diffusion of small ions and 2–3 kDa molecules, respectively) slows down dye equilibration (Supplementary Fig. 11). These examples, however, should be further constrained by experimental data, mainly because large molecules tend to undergo significant additional steric and viscous hindrance inside small cell compartments[70]. In contrast, smaller molecules can escape via gap junctions: this feature could be explored with ASTRO (Supplementary Note 1, ASTRO User Guide, pp. 27–28) once the gap junction diffusion sink rate has been experimentally constrained.

Finally, we explored simulation settings shown in Fig. 7 to ask how the clustering of $IP_3$-dependent $Ca^{2+}$ sources affects local $Ca^{2+}$ activity. It appears that spreading this signalling mechanism into individual (equally spaced) clusters, with the same total amount of the $IP_3$ activity, prompts de novo $Ca^{2+}$ events which could feature prominently in fluorescent recordings (Supplementary Fig. 12).

## Discussion

The present study sought to create a simulation tool ASTRO that would allow exploration and testing of mechanistic hypotheses pertinent to astroglial physiology, on the scale from nanoscopic processes to the entire cell expanse. Biophysical cell models replicating cell morphology have significantly influenced our understanding of neural function yet there have hitherto been no similar tools available to study astroglia. We therefore aimed at filling this knowledge gap.

The task of recreating astroglia in silico included three steps. The first step was to construct a 'stem tree' based on experimental data on main astrocyte branches that are readily identifiable in the optical microscope. This procedure is similar to the common 3D reconstructions of nerve cells using z-stacks of their optical sections. The second step, which was the key methodological challenge here, was to recreate the complex morphology of numerous nanoscopic astroglial processes pervading the synaptic neuropil. We therefore developed algorithms and computational tools (a) to quantify such processes using an empirical 3D EM sample and (b) to transform them into NEURON-compatible (cylinder-compartment based) shapes with matching biophysical properties. The latter was to be verified using dedicated Monte Carlo simulation tests (for diffusion and electrodiffusion) incorporated in ASTRO. This procedure provided all the key statistics characterising nanoscopic astroglial processes in the model.

Thus, the third step was to populate the modelled stem tree with stochastically generated nanoscopic processes, in accord with their morphometric stats obtained as outlined above. The main experimental constraint here was the VF occupied by astroglial processes, which we and others could measure directly using either 2PE microscopy or 3D EM. Because neighbouring astrocytes do not overlap in tissue, local VF faithfully reflects the space-filling properties of individual astroglial cells. We could therefore stochastically generate individual nanoscopic processes on dendritic branches until their bulk matched the empirical VF distribution. This procedure was to complete the modelled cell architecture: the model could now be explored using NEURON simulation environment which we equipped with several additional functions specific to astroglia.

In our case study, we obtained detailed morphological data on hippocampal astroglia in area CA1 using 3D EM and 2PE imaging, recreated the 'typical' cell in silico, and partly constrained its functional features by further experimental tests. Exploring the model has shed light on some traits of astroglial physiology that have not been attainable in experiments. Our simulations predict that local glutamate uptake or $K^+$ intake via $K_{ir}4.1$ generate only

very small membrane depolarisation across the astrocyte. It appears that transient rises of extracellular $K^+$ concentration prompt relatively small changes of intra-astroglial $K^+$, which dissipate relatively quickly, within one cell, due to efficient $K^+$ efflux through $K_{ir}4.1$ channels. Our tests illustrated that relatively small changes in $Ca^{2+}$ buffering properties might significantly influence the spread of regenerative intra-glial $Ca^{2+}$ waves. The model also showed that the classical mechanisms of rapid $Ca^{2+}$ sparks and hotspots described in other cell types could be consistent with the slow $Ca^{2+}$ signals reported by common $Ca^{2+}$-sensitive fluorescent indicators.

Thus, the modelling approach presented here serves several general purposes. Firstly, to recapitulate complex astrocyte morphology at multiple scales, paving the way to interrogating the form-function relationship in astroglia. Secondly, to assess whether a certain interpretation of experimental observations in astroglia is biophysically plausible. Thirdly, to understand the microscopic spatiotemporal dynamics of ion currents, molecular fluxes, and chemical reactions that cannot be monitored or registered experimentally. Finally, to predict the relationships between specific cellular features (morphology, $Ca^{2+}$ buffering, channel current density, molecular transport, etc.) and the physiological phenotype registered experimentally.

ASTRO can employ all synaptic and non-synaptic receptor mechanisms enabled by NEURON, thus enabling simulations with arbitrary patterns of network influences on the modelled cell. In addition, excitatory synaptic function could be mimicked using the glutamate uncaging ASTRO-NEURON menu. Nonetheless, exploring any receptor action specific (be this $IP_3$ release, $Ca^{2+}$ entry, $K^+$ fluxes, etc.) will require a dedicated study in which receptor kinetics and expression pattern are constrained, at least in part, by experimental tests.

Finally, we stress that our aim was not to present a fixed astrocyte model. Instead, we sought to create a flexible model builder ASTRO that would enable researchers to test biophysical causality of their experimental observations in various astroglial types. The examples presented here illustrate how such tasks could be accomplished. Likewise, ASTRO itself is not a fixed tool: as new hypotheses and investigatory tasks emerge, it will be upgraded and equipped with additional modelling features. The current version of ASTRO is accessible for download and exploration at https://github.com/LeonidSavtchenko/Astro.

## Methods

### Experimental methods: electrophysiology ex vivo.
Acute hippocampal transverse slices (350 μm thick) were prepared from P21–28 Sprague-Dawley rats, in full compliance with the national guidelines, the European Communities Council Directive of 24 November 1986 and the European Directive 2010/63/EU on the Protection of Animals used for Scientific Purposes, with the protocols approved by the UK Home Office. Slices were prepared in an ice-cold slicing solution containing (in mM): NaCl 60, sucrose 105, $NaHCO_3$ 26, KCl 2.5, $NaH_2PO_4$ 1.25, $MgCl_2$ 7, $CaCl_2$ 0.5, glucose 11, ascorbic acid 1.3 and sodium pyruvate 3 (osmolarity 300–310 mOsM), stored in the slicing solution at 34 °C for 15 min and transferred for storage in an extracellular solution containing (in mM): NaCl 125, $NaHCO_3$ 26, KCl 2.5, $NaH_2PO_4$ 1.25, $MgSO_4$ 1.3, $CaCl_2$ 2 and glucose 16 (osmolarity 300–305 mOsm). All solutions were continuously bubbled with 95% O2/5% CO2. Slices were allowed to rest for at least 60 min before recordings started.

Whole-cell patch-clamp recordings of stratum radiatum astroglia were performed in a submersion-type recording chamber. Slices were superfused with an extracellular solution containing (in mM): NaCl 125, $NaHCO_3$ 26, KCl 2.5, $NaH_2PO_4$ 1.25, $MgSO_4$ 1.3, $CaCl_2$ 2 and glucose 16 (osmolarity 300–305 mOsm), continuously bubbled with 95% O2/5% CO2. Whole-cell recordings were obtained with patch pipettes (3–5 MΩ) with an intracellular solution containing (in mM): $KCH_3O_3S$ 135, HEPES 10, Tris-phosphocreatine 10, $MgCl_2$ 4, Na2ATP 4, Na3GTP 0.4 (pH adjusted to 7.2 with KOH, osmolarity 290–295 mOsM). The cell-impermeable $Ca^{2+}$ indicator OGB-1 (200 μM unless indicated otherwise; Invitrogen O6806) was added to the internal solution. CA1 . protoplasmic astrocytes located in the stratum radiatum were identified either by full visualisation in fluorescence mode or in a DIC mode by their small soma size, low

resting potential (below −80 mV) and low input resistance (<10 MΩ). Astrocytes were held in voltage clamp at their resting potential or in current clamp.

In some experiments, whole-cell patches were excised by pulling gently the patch pipette attached to the cell body until the patch was completely detached from the processes and its membrane sealed. Estimation of the patch capacitance $c$ was carried out using a classical voltage-step method in which (a) a brief voltage step $\Delta V$ is applied, (b) the area under the transient capacitance current after the end of the voltage step is measured, giving electric charge $Q$, and (c) membrane patch capacitance is estimated as $c = Q/\Delta V$. Thus the patch area is evaluated from the ratio $c / c_m$ where specific membrane capacitance $c_m = 1$ μF $cm^{-2}$ is a common characteristic of astroglial membranes[39]. The steady-state current response to the voltage step was used to calculate the patch conductance, which was then normalized to the membrane area to obtain $G_m$.

### Experimental methods: 2PE imaging, uncaging, and FRAP ex vivo.
Astrocytes were filled via whole-cell patch clamp with 40–100 μM Alexa Fluor 594 for 15–20 min, as described previously[20,26,56]. We used an Olympus FV1000 imaging system optically linked to a femtosecond pulse Ti-sapphire MaiTai laser (Newport Spectra-physics). Cells were imaged using an Olympus XLPlan N 25x water immersion objective. Fluorescence recordings were obtained in line-scan mode (500 Hz, line placed through the astrocyte arbour and across the soma) at $\lambda = 800$ nm at an increased laser power of 15–20 mW under the objective to induce substantial bleaching of Alexa Fluor 594. Fluorescence was collected for 750–1000 ms, then a mechanical shutter was placed in front of the laser beam for 1–2 s to allow fluorescence recovery[20,26,56].

We used a combined two-photon uncaging and imaging microscope (Olympus, FV-1000MPE) powered by two Ti:Sapphire pulsed lasers (Chameleon, Coherent, tuned to 720 nm for uncaging and MaiTai, Spectra Physics, tuned to 800 nm for imaging) or, a Femto2D microscope (Femtonics, Budapest) coupled with two MaiTai lasers, and fully integrated with patch-clamp. Cells were imaged using an Olympus XLPlan N ×25 water immersion objective. The intensity of the imaging and uncaging laser beams under the objective was set to ~5 and 12–17 mW, respectively. Fluorescence recordings were normally carried out in line-scan mode (500 Hz); in FRAP experiments, laser power was increased to 15-20 mW during photobleaching epochs.

To record spontaneous $Ca^{2+}$ transients in frame scan mode, 200 μM Fluo-4 (Invitrogen) and 100 μM Alexa Fluor 594 (Invitrogen) were added to the intracellular solution. 350–500 $\mu m^2$ fields of view where imaged within the arbour of the patched astrocyte and the fluorescence emitted by Alexa Fluor 594 and Fluo-4 was collected at a rate of 3–5 Hz for 2–3 min. Time-dependent fluorescence transients were expressed as $\Delta G/R$ where $G$ corresponds to the background-subtracted Fluo-4 fluorescence and $R$ to the background-subtracted Alexa Fluor 594 fluorescence. Further details of the imaging methods were reported previously[20,26,56].

For MNI-glutamate uncaging, astrocytes were loaded with 100 μM Alexa Fluor 594 as a morphological marker. Astrocytes were held in voltage-clamp mode at their resting membrane potential (typically between −80 and −90 mV). The MNI-glutamate (12.5 mM) was either puffed within the tissue from a glass pipette placed above the patched cell, or added to the bath at 2.5 mM. Glutamate was uncaged for 20 ms at different distances from the soma (5–25 μm).

### Experimental methods: 3D reconstruction of live astrocyte stem tree.
A stratum radiatum astrocyte was held in whole-cell mode, with Alexa Fluor 594 added to the intracellular solution (see above; excitation at $\lambda_x^{2P} = 800$ nm). A z-stack of 2PE images was collected using $100 \times 100$ μm (512 × 512 pixel) individual frames containing the entire visible astrocyte structure, with a 0.5 μm z-step over 61 μm in depth. The image stack was stored (8-bit tiff format), individual images were corrected for the depth-dependent, quasi-exponential fluorescence signal decrease (Fiji Image-Adjust-Bleach Correction, plugin by Kota Miura 2014: 10.5281/zenodo.30769). Fluorescence background was subtracted (Fiji Image-Process), identifiable cell branches (>0.3–0.5 μm in diameter) were traced semi-automatically in individual 2D optical sections and reconstructed in 3D using Neurite Tracer (Fiji Plugins-Segmentation-Simple Neurite Tracer; by Mark Longhair and Tiago Ferreira, MRC and Janelia Campus; http://imagej.net/Simple_Neurite_Tracer; default segmentation sigma, 0.196). The data sets representing diameters of tubular compartments and their 3D co-ordinates (pairs of end points) were stores in SWC format. The Vaa3D software (Allen Institute, http://www.alleninstitute.org/what-we-do/brain-science/research/products-tools/vaa3d/) was used to convert these data sets into NEURON compatible files providing 3D structure of the astroglia stem-tree (with tubular compartments representing individual cylindrical compartments).

### Experimental methods: astroglia-targeted expression of GCaMP6f in vivo.
Animal procedures were conducted in accordance with the European Commission Directive (86/609/ EEC) and the United Kingdom (Scientific Procedures) Act (1986), with the protocols approved by the UK Home Office. Young male C57BL/6 mice (2–3 weeks of age) were anaesthetized using isoflurane (5% induction, 1.5-2.5% v/v). Subcutaneous analgesic (buprenorphine, 60 μg $kg^{-1}$) was administered and the animal was secured in a stereotaxic frame (David Kopf Instruments,

CA, USA) and kept warm on a heating blanket. The scalp was shaved and disinfected using three washes of topical chlorhexidine. Upon loss of pedal withdrawal reflexes, a small midline incision was made to expose the skull. A craniotomy of ~1–2 mm diameter was performed over the right somatosensory cortical region using a high-speed hand drill (Proxxon, Föhren, Germany). Stereotactic coordinates were +0.1 mm on the anterioposterior axis relative to bregma, and 2 mm lateral to midline. Once exposed, a warmed aCSF variant (cortex buffer, in mM; 125 NaCl, 2.5 KCl, 10 HEPES, 10 glucose, 2 CaCl₂, 2 MgSO₄) was applied to the skull and cortical surface throughout the procedure.

AAV5 *GfaABC1D-LckGCaMP6f* (Penn Vector Core, PA, USA) was pressure injected into the somatosensory cortex using a pulled glass micropipette stereotactically guided to a depth of 0.6 mm beneath the pial surface, at a rate of ~1 nL s⁻¹. A given injection bolus contained between 0.25 and $0.5 \times 10^{10}$ genomic copies, in a volume not exceeding 500 nL. After injection, pipettes were left in place for 5 min before retraction. The scalp was sutured with absorbable 7–0 sutures (Ethicon Endo-Surgery GmbH, Norderstedt, Germany) and the animal was left to recover in a heated chamber. Meloxicam (subcutaneous, 1 mg kg⁻¹) was administered once daily for up to two days following surgery. After a 4–6-week AAV incubation period, animals were prepared for multiphoton imaging through a cranial window implantation as described below.

### Experimental methods: two-photon excitation imaging of astroglia in vivo.

Following viral transduction of LckGCaMP6f as above, male C57BL/6 mice (7–9 weeks of age) were prepared for cranial window implantation and 2PE microscopy. Animals were anaesthetized using fentanyl, midazolam and medetomidine (i.p., 0.05, 5 and 0.5 mg kg⁻¹, respectively). Adequate anaesthesia was ensured by continuously checking for the loss of pedal withdrawal reflexes and anaesthesia was supplemented appropriately throughout the procedure (typically 10–20 % of the original dose per hour). Body temperature was maintained at 37.0 ± 0.5 °C using a feedback rectal thermometer and heating blanket. The animal was secured in a stereotaxic frame and a craniotomy of ~2.5 mm diameter was carried out over the right somatosensory cortex, centred 0.2 mm caudal to bregma and ~2.5 mm laterally from the midline. Once exposed, the cortical surface was continuously superfused with warmed aCSF (in mM; 125 NaCl, 2.5 KCl, 26 NaHCO₃, 1.25 Na₂HPO₄,18 Glucose, 2 CaCl₂, 2 MgSO₄; saturated with 95% O₂/5% CO₂, pH 7.4). Cortical astrocytes were labelled using multicell bolus loading of sulforhodamine 101 (SR101, 5 μM; SR101 (in cortex buffer vehicle) was pressure-injected through a pulled glass micropipette targeted to 2–3 injection sites within the transduced region, comprising a total volume of 500 nL. The cortical surface was covered with 1% agarose and a glass coverslip was placed on top. Using tissue adhesive (Dermafuse, Vet-Tech Solutions, UK), the coverslip was partially secured and a custom-built headplate fixed to the skull. A single cranial-mounted screw was inserted over the contralateral hemisphere and the entire assembly was then secured using dental cement. During imaging, the headplate was used to secure the animal under the objective on a custom-built stage.

In these experiments, two-photon excitation was carried out using a Newport-Spectraphysics Ti:sapphire MaiTai laser pulsing at 80 MHz, and an Olympus FV1000 with XLPlan N ×25 water immersion objective (NA 1.05). Acquisitions were carried out using a wavelength of 920 nm and the mean laser power under the objective was kept at 20–35 mW. Cortical astrocytes were readily identified through SR101 labelling and verified for GCaMP6f expression by frame-scanning for calcium transient activity. Recordings were made at a depth between 50 and 250 μm from the cortical surface. XY time series (at 0.5–2 Hz with a pixel dwell time of 0.5–4 μs and pixel size of 0.248–1.59 μm) were taken in identified regions to measure spontaneous calcium activity.

### Experimental methods: fast fixation and DAB staining of recorded astrocytes.

In a subset of experiments we loaded an astrocyte with biocytin, and after the experiment the slices were rapidly fixed (by submersion) with 1.25% glutaraldehyde and 2.5% paraformaldehyde in 0.1 M PB (phosphate buffer, pH 7.4), to be kept overnight, submerged in 10% sucrose in PB for 10 min and then in 20% sucrose in PB for 30 min. The slices were consequentially freeze-thawed in liquid freon and liquid nitrogen for 3 s each to gently crack intracellular membranes and embedded in 1% low gelling temperature agarose in PB (Sigma-Aldrich, USA). Embedded slices were sectioned at 50 μm on a vibrating microtome (VT1000; Leica, Milton Keynes, UK). Sections (50 μm) sections were incubated in 1% H₂O₂ in PB for 20 min to eliminate blood background, washed with 0.1 M TBS (tris buffer saline, pH 7.4) and incubated with ABC solution (VECTASTAIN ABC, Vector laboratories, USA) for 30 min at room temperature. Next sections were washed with 0.1 M TB (tris buffer, pH 7.4), pre-incubated with DAB (3,3′-Diaminobenzidine tablets—Sigma-Aldrich, USA) solution (10 mg DAB tablet + 40 ml TB) for 30 min at room temperature in dark and finally incubated with DAB + H₂O₂ solution (5 μl of 33% H₂O₂ + 25 ml of DAB solution) for 10–20 min at room temperature in the dark. The DAB stained sections was washed in PB, post-fixed in 2% osmium tetroxide and further processing and embedding protocols were essentially similar to those reported previously[26]. Briefly, the tissue was dehydrated in graded aqueous solutions of ethanol (40–100%) followed by 3 times in 100% acetone, embedded into a mixture of 50% epoxy resin (Epon 812/Araldite M) and 50% acetone for 30 min at room temperature, embedded in pure epoxy resin, and polymerized overnight at

80 °C. Sections in blocks were coded and all further analyses were carried out blind as to the experimental status of the tissue.

### Experimental methods: 3D electron microscopy.

Serial sections (60–70 nm thick) were cut with a Diatome diamond knife and systematically collected using Pioloform-coated slot copper grids (each series consist of up to 100 serial sections). Sections were counterstained with 4% uranyl acetate, followed by lead citrate. Finally sections were imaged in stratum radiatum area of CA1 (hippocampus) using AMT XR60 12 megapixel camera in JEOL 1400 electron microscope. Serial sections were aligned as JPEG images using SEM align 1.26b (software available from http://synapses.clm.utexas.edu/). 3D reconstructions of DAB-stained astrocyte fragments and the adjoined to stained astrocytes dendritic spines (that host clearly identifiable excitatory synapses) were performed in Trace 1.6b software (http://synapses.clm.utexas.edu/). 3D reconstructions of selected astrocytic segments and dendritic spines were imported to 3D-Studio-Max 8 software for rendering of the reconstructed structures. However, ASTRO can upload other standard text-formatted files with 3D coordinates representing the cell-membrane 'mesh'.

### Statistics summary.

The present study contained no longitudinal or multifactorial experimental designs. In electrophysiological or imaging experiments the main source of biological variance was either individual cells or individual preparations (the latter in case of field measurements in acute slices), as indicated. In accord with established practice, in the ex vivo tests we routinely used one cell per slice per animal, which thus constituted equivalent statistical units in the context of sampling, unless indicated otherwise. Statistical hypotheses pertinent to mean comparisons were tested using a standard two-tailed *t*-test, unless the sample showed a significant deviation from Normality, in which case non-parametric tests were used as indicated. The null-hypothesis rejection-level was set at $\alpha = 0.05$, and the statistical power was monitored to ensure that that the sample size and variance were adequate to detect a mean difference (in two-sample comparisons) of 10–15% or less.

### Astrocyte model: generating 'invisible' nanoscopic morphology.

Nanoscopic processes of the astrocyte model were generated in a probabilistic manner based on the sample statistics from 3D EM reconstructions (Fig. 2). The total cell surface area $S_{tot}$ represented by the cylinder-based shape approximations (Fig. 2d, e), consists of the (lateral) surface areas of all cylinder-compartment sides $S_{lat}$ added to the surface areas of 'main' cylinder bases $S_M$ (blue in Fig. 2d, bottom) minus the surface areas of 'transitional' cylinder bases $S_T$ (green in Fig. 2d, bottom). In our case study, computations indicated that $S_T = 0.20 S_M$ throughout modelling: thus, the formula $S_{tot} = S_{lat} + 0.8 S_M$ was applied. In the generated cell models, the S/V ratios were ranging from ~7 μm⁻¹ near the soma to an average of ~22 μm⁻¹ in the bulk of the cell arbour, in accord with the empirical observations.

### Astrocyte model: transporter/channel kinetics and diffusion-reaction mechanisms.

Models built with ASTRO can incorporate many dozens of NEURON-enabled channel and transporter kinetic mechanisms that have been tested and validated in numerous studies combining experiments and simulations[10]. The formal descriptions of the respective algorithms could be found using an extensive NEURON database at https://senselab.med.yale.edu/modeldb/ which also contains references and links to the original studies and the mathematical formulism involved. Upon ASTRO installation on the host computer, these mechanisms could also be inspected in the respective *.mod files in the 'neuronsims' directory or, alternatively, online here https://github.com/LeonidSavtchenko/Astro/tree/master/neuronSims.

Several channel current and diffusion-reaction mechanisms have been written specifically for the present model. The kinetics of glutamate transporter GLT-1 involving glutamate and ion fluxes has been incorporated in accordance with[43] (description in the GluTrans.mod file). The $K_{ir}$4.1 potassium current has been incorporated in accordance with[17] (Kir4.mod; Fig. 5 legend), intracellular K⁺ diffusion was incorporated as longitudinal diffusion (no radial rings) using a built-in Ca²⁺ diffusion algorithm described in the next section (potassium.mod), the FRAP mechanism incorporated the same algorithm plus a reaction-diffusion step (FRAP.mod), and K⁺ extrusion was modelled as a first-order pump (kpump.mod; Supplementary Fig. 7 legend). Gap junction mechanisms were modelled either as a (zero-order) current leak (gap.mod) or as a diffuse escape (gapCa.mod). Throughout these mechanisms, the respective kinetic parameters can be set using the relevant NEURON-enabled ASTRO menus, as described in the User Guide (https://github.com/LeonidSavtchenko/Astro/blob/master/ASTRO_User_Guide.pdf).

### Astrocyte model: Ca²⁺ homoeostasis and diffusion.

ASTRO simulation algorithms enabling intracellular Ca²⁺ homoeostasis and diffusion (including that among adjacent compartments of unequal size) are detailed in Chapter 9 of the NEURON Book[10] (also here https://www.neuron.yale.edu/neuron/docs), and can be found in the modified cadifus.mod file in the model installation. In brief, Ca²⁺ diffuses freely whereas buffer-bound Ca²⁺ (which has much lower diffusivity) is considered stationary, for the sake of simplicity. In individual cylindrical cell

compartments, radial diffusion occurs through four concentric shells surrounding a cylindrical central core, and longitudinal diffusion is calculated using fluxes between the corresponding concentric compartments adjusted for the cross-section areas. The longitudinal and radial diffusion coefficient for $Ca^{2+}$ was set to 0.3 $\mu m^2$ $ms^{-1}$, the basal level was set to 50 nM, and $IP_3$ concentration at 0 unless specified otherwise.

In addition to free diffusion, $Ca^{2+}$ homoeostasis mechanisms included the SERCA pump, SERCA channel and SERCA leak, the endogenous (stationary) and exogenous ($Ca^{2+}$ indicator) mobile buffers, and a plasma membrane $Ca^{2+}$ pump with the threshold mechanism (cadifus.mod). The kinetics of buffers can be modified using NEURON-enabled menus. The mechanistic details of $Ca^{2+}$ SERCA pump were as described earlier[71]. The current model implementation assumes that $IP_3$ is distributed uniformly across cell compartments, i.e. that diffusion equilibration of $IP_3$ is fast compared to $Ca^{2+}$ concentration transients in space or time.

**Modelling with ASTRO: on-line access and installation**. Detailed information on the installation and running of ASTRO can be found in the User's Manual (Supplementary Material; online download at https://github.com/ LeonidSavtchenko/Astro/blob/master/Manual). The current version of ASTRO can also be downloaded directly from https://github.com/LeonidSavtchenko/Astro. The (regularly updated) User Guide can be downloaded from the same location or found (current version) in the Supplementary Note.

In brief, running ASTRO with de novo 3D EM reconstructions and model building, but without full-scale simulations of intracellular $Ca^{2+}$ dynamics, requires the host computer to have MATLAB (2012 version or later, https://uk.mathworks. com/products/matlab.html), or at least free-download MATLAB Runtime package (https://www.mathworks.com/priducts/compiler/mcr), and NEURON (7.2 or later, https://neuron.yale.edu/neuron/download) installed under Windows 7 or Windows 10. Simuations using ready-made astrocyte models require NEURON installation only.

Simulating full intracellular $Ca^{2+}$ dynamics is highly resource-consuming and normally requires an additional Worker computer/cluster operating under Linux, with preinstalled NEURON (https://neuron.yale.edu/neuron/download/ compile_linux) and MPI whereas the Host computer will require MATLAB (2013 version or later), NEURON (7.0 or later), and access to the Internet. In house, the Linux version with the parallel computations including intracellular diffusion simulation was routinely run using a 12-node in-house computer cluster[72], taking advantage of the computational optimization routines developed by us earlier for compartmental models and Monte Carlo simulations[30,31,72].

## Data availability

The current version of ASTRO, with open code access, can also be downloaded directly from https://github.com/LeonidSavtchenko/Astro, with the User's Manual available at https://github.com/LeonidSavtchenko/Astro/blob/master/Manual). See above for further detail. All experimental recording data are available from the authors upon request.

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

## Acknowledgements

This study was supported by Wellcome Trust Principal Fellowship (101896 and 212251/Z/18/Z), European Research Council Advanced Grant (323113-NETSIGNAL), European Commission FP7 ITN Extrabrain (606950 EXTRABRAIN), Russian Science Foundation grant 15-14-30000 (Fig. 3A-B data) (D.A.R.); German Research Foundation (DFG, SFB1089 B03, SPP1757 HE6949/1 and HE6949/3), the European Commission ITN EU-Glia, Human Frontiers Science Program, NRW Rückkehrerprogramm, UCL Excellence Bridging Award (C.H.). The authors thank Sergey Alexin and Volodymyr Hromakov (AMCBridge LLC) for inspirational help with software solutions.

## Author contributions

D.A.R., L.P.S. and C.H. conceived the study; L.P.S. developed and implemented the modelling approach and its computing environment; C.H. designed and carried out morphometric studies and analyses ex vivo; C.H. and L.B. carried out patch-clamp and imaging experiments and analyses ex vivo; T.J. carried out some imaging experiments ex vivo; J.P.R. carried out imaging experiments in vivo; I.K., M.M. and M.G.S. designed and carried out quantitative 3D EM studies; D.A.R. narrated the study, carried out 3D stem-tree reconstructions, some data analyses and wrote the paper which was subsequently contributed to by all the authors.Data availabilityThe current version of ASTRO, with open code access, can also be downloaded directly from https://github.com/LeonidSavtchenko/Astro, with the User's Manual available at https://github.com/LeonidSavtchenko/Astro/blob/master/Manual. See above for further detail. All experimental recording data are available from the authors upon request.

## Additional information

**Competing interests:** The authors declare no competing interests.

