## [Peer Review File · Nature Communications]

Reviewers' comments:

Reviewer #1 (Remarks to the Author):

A major limitation to our understanding of the function of astrocytes is the lack of a realistic computational model of the physiological properties of astrocytes. In this paper the authors have undertaken an impressive project with a grand goal, to generate an astrocyte in silico model that can be tested and ultimately used to delineate astrocyte functions. This model will be of great use for neuroscientists investigating the functions of astrocytes in the brain.

The shortcoming of such an approach is that we don't fully understand the distribution of ion channels, transporter etc at distances from the soma where voltage clamp control is impossible. However the utility of this approach should be the flexibility to incorporate new advances and discoveries to further refine the model.

The authors used a reasonable strategy to create this model. This involved: 1) reproducing the morphological structure mathematically based on two-photon imaging of Alexa Fluor 594 initially based on the principal branch structure (those whose thickness was above the diffraction limit, $\sim 0.4 \mu\text{m}$); 2) secondarily including mathematical representations of "irregularly shaped nanoscopic processes" that can only be defined using electron microscopy. The nanoscopic processes were examined using careful EM analysis of DAB photoconversion of biocytin injected astrocytes. The volumes of the astrocytes were preserved (hopefully) by rapid fixation of this slices by submersion. There are issues with truly understanding the shape and volume of astrocytes within the brain because of the impact of every fixation or measurement protocol. However, the authors provide an impressive and scholarly discussion of the different approaches they took and in the end I think they have done the morphological reproduction as best as is possible using today's technology. In addition, as the authors point out "The sampling and transformation procedure for nanoscopic processes has been integrated in ASTRO for investigators to explore, monitor, and validate" (page 7). Therefore, future readers and users of the software can investigate these issues using the software.

The tissue volume fraction section is quite important as the authors mention for understanding a number of potential functions of astrocytes (page7). The authors use an interesting strategy to estimate the tissue volume fraction by comparing the fluorescence intensity at the astrocyte soma (100% tissue volume at that point) to the fluorescence signal intensity of the adjacent regions with astrocyte processes where the processes represent a fraction of the volume. The authors should discuss and is possible estimate the error that might arise from this strategy that would be due to the point-spread-function of their imaging system. This would arise from the detection of fluorescence from multiple submicron processes that might be within the blurred spread in the Z-axis.

The section on modelling the passive electrical properties is quite clear and well done. The authors could comment on how their model would be altered if there were uneven distributions of K channels open at rest etc that could lead to variations in membrane conductance in the glial processes. Some experimental data have suggested that the whole

cell voltage clamp recordings are only controlling a small region of the astrocyte around the soma. Any experiments on astrocyte that provide any indications of the distribution of K channels throughout different regions within astrocytes would be very useful at this point in the paper. In any case the comparison of the voltage spread in the modelled astrocyte is a powerful result.

The section on intracellular calcium modelling will be very useful for the astrocyte field. However there is a shortcoming in the discussion as the models as significant spontaneous calcium transients have been reported in the IP3R2-/- knock mouse (Haustein et al, 2014; Rungta et al 2016; Agarwal et al 2017). This should at least be mentioned and included in their discussion of sources of calcium signaling in the generation of local signals.

In summary this is a well written manuscript providing a mathematical model that is well justified and carefully described. This will be a very useful tool for scientists studying astrocytes.

Reviewer #2 (Remarks to the Author):

The authors of the manuscript "Disentangling astroglial physiology with a realistic cell model" have undertaken the daunting task of an in silico modeling of an astrocyte cell. The resulting model is implemented in a computer program using the NEURON interface, which allows manipulation of basic parameters and monitoring the effects on the biophysical properties of the astrocyte. This program is an important tool, as one can use it to test various hypothesis and gain better understanding of the biophysical properties of astrocytes and may be of interest to glial biologists, neurophysiologists, and computational biologists. However, the article are a few specific concerns detailed below.

Major

1. The outline and flow of the manuscript is logical but each section is lacking sufficient detail and it is difficult to ascertain what exactly was done and why.
2. The explanation of the model is lacking. The framework of the model is not well described, no equations are provided, the parameters used are not always provided, and their general stability has not been tested, i.e. how changing the measured or assumed parameters affect the simulations. E.g., how does changing the branch number within the estimation of data scatter (4-9 branches) affects the simulations? How does changing the tissue volume fraction within the standard error (7-13% according to Figure 3b) affect the morphology of the cell and its biophysical properties, etc.
3. It would be useful to use to test the model by testing a few predictions. For example:
 - a. How does the model predict the diffusion of larger molecules with lower diffusion coefficient such as dextran-conjugated alexa?
 - b. How does blocking potassium channel block affects the current of glutamate uncaging?
 - c. How does change in astrocytic volume affect its biophysical properties?
 - d. How does block of addition of intracellular calcium buffer affects the propagation of calcium waves?
 - e. Etc.

4. The model is based on an isolated astrocyte. This is, of course, is not the case for astrocytes in the hippocampus or most other brain regions which show a high degree of coupling to one another. Indeed, the coupling of astrocytes likely influences most of the biophysical properties this program attempts to address. This parameter is not really considered in the new model.

Specific comments

1. The title of the manuscript is "Disentangling astroglial physiology with a realistic cell model". However, the model described mainly deals with the biophysical properties of the astrocyte, and not with its physiological function (i.e. extracellular potassium and neurotransmitter buffering, blood-brain barrier maintenance, metabolic support of neurons etc.). A title such as "Disentangling astroglial biophysical properties with a realistic cell model" is more suitable.
2. The introduction lacks any summary of previous attempts to model astrocytes.
3. The "Reconstructing astrocyte gross morphology" section in the Results is missing important information which makes the readers interpretation difficult. i.e Why were several programs were used for the 3D reconstruction? Was one better than the others? Did they converge? What parameters were extracted and used in the model? Which of the approaches were used for stem tree construction? This section needs to be re-written more clearly.
4. More detail is needed to understand how the 'transitional' slab cross section was calculated.
5. The authors should more clearly define the difference between stem branches and nanoscopic protrusions. It is not clear how the nanoscopic processes were added to the stem tree of the astrocytes, and how it was later adjusted to fit the calculation of volume fraction and surface to volume ration. In addition, as above, it is not clear whether these parameters are stable, or a slightly different parameters could lead to a significant change in the morphology of the model astrocyte.
6. In Figure 3, what is the diffusion coefficient used for the modelling of FRAP? Also, the FRAP did not seem to recover much, if at all. Perhaps longer recovery times are needed. In Figure 3g, it is not clear what is normalized FRAP time. Is it the τ of recovery? Also, the stability of the rate of Alexa line-scan photobleaching is not tested for stability.
7. The "Passive electrical properties of astrocytes" (Figure 4a-c) is poorly explained, especially how the G_m parameter estimate was obtained.
8. In the section "Membrane voltage landscape generated by local glutamate uptake", it is not clear how GLT1 were "scattered". Also, specific GLT1 kinetics used here should be noted.
9. In the section "Potassium uptake and redistribution inside astrocytes", the kinetics and parameters of the first order pump used for mimicking potassium intrusion should be noted in text, as well an explanation in text of the parameter j_{ext} . Units of j_{ext} should be noted in Figure.
10. The modelling uses in vitro parameters of CA1 astrocytes. However, modeling of calcium waves uses in vivo measurements of somatosensory cortical astrocytes. A more appropriate way to do this is to measure calcium waves in ex vivo CA1 astrocytes.
11. The authors should add to their discussion several topics: what is this model good for, and what are its weaknesses? How one can extract physiological effects from this

biophysical model? How can this model be further developed (e.g. addition of gap junctions, end feet, changes in volume, changes in synaptic coverage)? How is the tripartite synapse is represented in the model?

Minor comments:

1. In many figures it would be easier to understand if there was an accompanying supplementary movie of the modelling, and not just a snapshot: e.g. Figure 2e, Figure 7
2. What is the red box in Figure 1e?
3. Figure 2b has bad colors, especially yellow on white background.
4. How thick are the slices in Figure 2c?
5. What are the different shades of blue in Figure 2d?
6. In Figure 2f, at which plane where the particles release, and where was the diffusion flux measured? Also, the use of ampere instead of $e\text{ ms}^{-1}$ (use electrons instead of generic charged particle) is easier to comprehend. Add electrical parameters to Figure legend as in Supplementary Figure 3.
7. Where is representative example in Figure 3f measured?
8. Figure 4 Y axis title should be "Input" not "Inpur".
9. In Figure 6b, scale is not provided, especially important in the lower panel (" dC/dt ").
10. In page 8 there are references to Supplementary Information. However, only Supplementary Figures were provided.
11. Page 11 second paragraph 5th sentence, should be "Kir4.1" nor "Kir1.4".
12. In Methods, electrophysiology ex vivo section, last paragraph, 7th sentence there are two repetitive words: "Ascorbic ascorbic..." and "Sodium sodium...".
13. In Methods, Fast fixation and DAB staining of recorded astrocytes, 4th sentence, it is not clear what "infiltrated" means"

Reviewer #3 (Remarks to the Author):

This study uses the NEURON computational environment to model basic properties of astrocytes. The report is well written and easy to read, but I have several concerns. The premise for the model is that several basic properties of astrocytes are neglected. Also, the modeling does not provide interesting questions, but strictly replicate the biology already reported. A specialized glia journal may be a better fit for this study.

Major critique:

- (1) Astrocytes are expressing a large number of gap junction that couple the fine astrocytic processes belonging to the same cell and to its neighbors together. Gap junctions are fully permeable to Ca^{2+} , IP_3 and small tracers. This fundamental feature of astrocytes that will affect Ca^{2+} signaling, K^+ buffering, glutamate uptake and more is neglected
- (2) Oscillatory or wavelike propagation of Ca^{2+} waves within single astrocytes is a sign of pathology. It is often observed in slices or in anesthetized mice after poor surgery or in response to excessive light stimulation. Awake behaving mice display synchronous increases in Ca^{2+} that is blocked by α_1 receptor antagonists. Why model data based on non-optimal preparation when this study is presented as a model of normal physiological

behavior?

(3) Astrocytes are highly polarized cells extending 1-3 vascular endfeet. These processes are a key to astrocytic function, but not included in the model. Why not?

(4) The model data are not providing any questions or prediction that would be interesting to test.

Reviewer #4 (Remarks to the Author):

The main achievement of the manuscript is developing a simulation tool for modeling calcium and potassium dynamics in a detailed astrocyte morphology of astrocytes. This software platform is shown to interphase well with 'Neuron', a popular neuron modeling tool that is used to simulate neuronal activity in realistic morphologies.

As far as I know, there is no extant tool box available that allows for modeling astrocytic calcium and potassium dynamics in realistic spatial domains. As a add-on functionality to 'Neuron', it will serve a large user base. Decoding fluorescent calcium dye measurements to calculate real calcium signal is an important contribution of this paradigm.

Substantial effort and time in the manuscript is spent on creating a canonical geometrical representations of an astrocyte that can be morphed into 'Neuron'. This required that 1) statistical description of the cellular microphysiology can be justified and modified to suit the user 2) ensure the cylinder-compartment discretization of space as required by 'Neuron' is a reasonable assumption that does not compromise accuracy. Here a exhaustive list of key morphological features and bounds on physiological parameters should have been made available. Also, the "MATLAB based algorithms" that are used in 'ASTRO' to generate nanoscopic processes is of potential scientific value, however not discussed at all. Unfortunately, once the model is ready, it falls short of any serious investigation for new insights. The authors carry out a few illustrative simulations in the model that describe intracellular potassium and calcium dynamics, but no interpretation or consolidated view on its significant is offered. Very little time is spend in exploring the various components of the model, model parameters are not studied systematically (except for buffering concentration and K_D).

A objective of a good realistic model allows you investigate and predict behavior under different stimuli and protocols and extend the understanding of experimental findings. This is precisely where the manuscript is lacking. New insights that could be obtained using this modeling paradigm and its functional implications to neuronal activity are not investigated at all.

Whereas experimental methods used to explore morphology have been detailed well and justified, the theoretical methods are not outlined. Unfortunately, it ends up being a software package wherein the algorithms used remain a black box.

Nature Communications may not be an appropriate platform for this manuscript in its present form since it does not represent a significant in sight in Neuroscience or Glial Biology. However 'ASTRO' developed here would be a useful add-on tool that would expand

the utility of Neuron to modeling astrocytes. The documentation and installation instructions are clearly written and users with some experience in 'Neuron' may be able to use it in a reasonable amount of time.

Here is list of more specific critique

1) Abstract itself does not reflect any new scientific insight but indicates that the tool box is the central achievement. For example "evaluate some of the basic features of astroglial physiology inaccessible in experiments such as membrane voltage propagation, K⁺ redistribution dynamics, or intracellular Ca²⁺ buffering capacity",

What is it that the evaluation leads to is not mentioned and its effect on function is not investigated. This is a crucial missing component throughout given that this is a modeling study. The abstract goes on to say "paves the way for mechanistic interpretations ..."

However the nature of these interpretations is not elaborated upon at all.

2) With all morphological measurements being carried out here, a quantification of a canonical astrocytic morphology with relevant statistical variation in a table should be shown.

3) Page 3, "We have therefore developed (MATLAB-based) computational algorithms and software that, firstly, use experimental data to recreate the space-filling architecture of nanoscopic astroglial processes, and, secondly, make this cell architecture NEURON-compatible."

These algorithms are not described.

4) Page 4, "which we have equipped with multiple additional functionalities specific to astroglia."

What additional functionalities specific to Neuroglia is not mentioned. Especially, the development of the toolbox is the main achievement here.

5) Page 10, "Equipped with these experimental constraints, we simulated a simple voltage-clamp experiment, which predicted that a voltage signal generated at the soma should spread

with an electrical 3D-space constant of 30-60 μm depending on the signal frequency (Fig. 4d, left). Intriguingly, a membrane voltage signal generated at the periphery was attenuated with a shorter space constant of 25-30 μm (Fig. 4d, right), the asymmetry previously reported in nerve cells"

Characterization of this 'intriguing' difference not investigated.

7) Page 10, "As the most likely scenario of astroglial depolarisation relates to the

accumulation of extracellular K^+ , these data should help to understand the electrogenic membrane mechanisms involved.”

The statement is too abstract. The mechanisms involved should be quantified using the developed model

8) There are more astrocyte specific models for IP3 and RyR with very different behavior. However off the shelf models from “Neuron” are used. This may be a problem for classification of calcium dynamics. A justification of the models used for receptors should be given as they can crucial change the results

9) Page 11, “Clearly, precise parameters of K^+ entry and extrusion will depend on the experimental circumstances that determine the kinetics of extracellular $[K^+]$, K^+ channel activity, and the voltage landscape of astroglial membrane 16. Once these and related quantities have been empirically constrained, ASTRO should provide important clues to the spatiotemporal dynamics of the intra-astroglial K^+ buffering and redistribution, on the scale from individual synapses to hundreds of microns.”

These precisely are the kind of computational experiments that can be carried out here to acquire a more complete understanding. A wide range of parameters for K^+ entry, extrusion and voltage landscape can be easily studied that would potentially provide a quantitative understanding complete range of potassium dynamics.

11) A systematic calculation of changes in calcium signaling with different clustering and inter-cluster distance of IP3 receptors and RyR using the model should be done

13) Effects of homogenous and heterogenous buffering should be classified

Biophysical underpinning of astroglial physiology probed with realistic cell models

Point-by-point reply to reviewers' comments

Reviewer #1 (Remarks to the Author):

We are grateful to the Reviewer for the careful analysis of our manuscript and the encouraging and insightful comments.

A major limitation to our understanding of the function of astrocytes is the lack of a realistic computational model of the physiological properties of astrocytes. In this paper the authors have undertaken an impressive project with a grand goal, to generate an astrocyte *in silico* model that can be tested and ultimately used to delineate astrocyte functions. This model will be of great use for neuroscientists investigating the functions of astrocytes in the brain.

The shortcoming of such an approach is that we don't fully understand the distribution of ion channels, transporter etc at distances from the soma where voltage clamp control is impossible. However the utility of this approach should be the flexibility to incorporate new advances and discoveries to further refine the model.

We thank the Reviewer for highlighting this key point. Indeed, ASTRO is not just a fixed astrocyte model: it is a highly flexible computational tool designed to help investigators to build and interrogate *in silico* various types of astroglia, in accord with their experimental observations and hypothesis testing.

The authors used a reasonable strategy to create this model. This involved: 1) reproducing the morphological structure mathematically based on two-photon imaging of Alexa Fluor 594 initially based on the principal branch structure (those whose thickness was above the diffraction limit, $\sim 0.4 \mu\text{m}$); 2) secondarily including mathematical representations of “irregularly shaped nanoscopic processes” that can only be defined using electron microscopy. The nanoscopic processes were examined using careful EM analysis of DAB photoconversion of biocytin injected astrocytes. The volumes of the astrocytes were preserved (hopefully) by rapid fixation of this slices by submersion. There are issues with truly understanding the shape and volume of astrocytes within the brain because of the impact of every fixation or measurement protocol. However, the authors provide an impressive and scholarly discussion of the different approaches they took and in the end I think they have done the morphological reproduction as best as is possible using today's technology. In addition, as the authors point out “The sampling and transformation procedure for nanoscopic processes has been integrated in ASTRO for investigators to explore, monitor, and validate” (page 7). Therefore, future readers and users of the software can investigate these issues using the software.

Correct.

The tissue volume fraction section is quite important as the authors mention for understanding a number of potential functions of astrocytes (page7). The authors use an interesting strategy to estimate the tissue volume fraction by comparing the fluorescence intensity at the astrocyte soma (100% tissue volume at that point) to the fluorescence signal intensity of the adjacent regions with astrocyte processes where the processes represent a fraction of the volume. The authors should discuss and is possible estimate the error that might arise from this strategy that would be due to the point-spread-function of their imaging system. This would arise from the detection of fluorescence from multiple submicron processes that might be within the blurred spread in the Z-axis.

Indeed, the point-spread-function (PSF) spans over $\sim 1 \mu\text{m}$ in the z direction and therefore will often fall across several fragments of neighbouring nanoscopic processes. Thus, the resulting emission intensity will reflect the local tissue volume fraction occupied by astroglia, namely by all local astroglial elements falling within that volume. We have added an explanatory illustration (new Supplementary Fig. 4) and expanded the text accordingly. In this context, our modelling

strategy was precisely to reproduce the astroglial VF distribution in tissue, accounting for neighbouring processes that might occur close to one another.

The section on modelling the passive electrical properties is quite clear and well done. The authors could comment on how their model would be altered if there were uneven distributions of K channels open at rest etc that could lead to variations in membrane conductance in the glial processes. Some experimental data have suggested that the whole cell voltage clamp recordings are only controlling a small region of the astrocyte around the soma.

These are important issues. ASTRO is equipped with NEURON-enabled, built-in mechanisms representing various potassium channels and a K^+ pump. In the revision, we have constructed and incorporated the $K_{ir}4.1$ channel mechanism^{1,2} into the settings, so that one could explore the relevant effects.

Indeed, unlike neurons, astrocytes are electrically leaky cells, mainly because they have an extraordinarily large plasma membrane area. This has two major consequences. Firstly, cell membrane depolarisation will require a very large current. Secondly, any locally generated current will stay local, with little influence on the membrane voltage landscape. For instance, spot-uncaged glutamate generates a significant transporter current hotspot but only a small voltage deflection away from the spot (revised Fig. 4f), and only tiny currents recorded at the soma (Fig. 4e). Thirdly, the unit conductance of $K_{ir}4.1$ in astrocytes is either compatible with or lower than the rest of membrane conductance^{1,2}. Therefore, adding more $K_{ir}4.1$ should have little effect on the passive voltage landscape (assuming that resting potential remains unchanged with additional $K_{ir}4.1$, see text). We have carried out additional tests to illustrate this (new Supplementary Fig. 6). Relevant explanations have been added to the text.

Any experiments on astrocyte that provide any indications of the distribution of K channels throughout different regions within astrocytes would be very useful at this point in the paper. In any case the comparison of the voltage spread in the modelled astrocyte is a powerful result.

We are not aware of the available tools to probe the subcellular distribution of functional astroglial K^+ channels. In our experience, local micropipette iontophoresis or pressure puffs of K^+ induced poorly controlled concomitants, such as mechanical interference. Nonetheless, we have incorporated the $K_{ir}4.1$ channel mechanism and the extracellular K^+ dynamics in the model and simulated various physiological scenarios (glutamate uncaging, extracellular K^+ rise) to understand their contributing roles (new Fig. 5; new Supplementary Fig. 6). While such data illustrate the role of $K_{ir}4.1$ channels (a major membrane mechanism in astroglia), further experimental constraints are required for the full appreciation of other contributors to membrane currents, such as ion exchangers, pumps, semi-channels, etc.

The section on intracellular calcium modelling will be very useful for the astrocyte field. However there is a shortcoming in the discussion as the models as significant spontaneous calcium transients have been reported in the $IP3R2^{-/-}$ knock mouse (Haustein et al, 2014; Rungta et al 2016; Agarwal et al 2017). This should at least be mentioned and included in their discussion of sources of calcium signaling in the generation of local signals.

We fully agree, and consistently highlighted that in our published reviews, that IP_3 receptors are not the only active source of Ca^{2+} signals in astroglia. We have expanded our explanations accordingly, citing the relevant papers as suggested.

In summary this is a well written manuscript providing a mathematical model that is well justified and carefully described. This will be a very useful tool for scientists studying astrocytes.

We are grateful to the Reviewer for the encouraging comments.

Reviewer #2 (Remarks to the Author):

The authors of the manuscript "Disentangling astroglial physiology with a realistic cell model" have undertaken the daunting task of an in silico modeling of an astrocyte cell. The resulting

model is implemented in a computer program using the NEURON interface, which allows manipulation of basic parameters and monitoring the effects on the biophysical properties of the astrocyte. This program is an important tool, as one can use it to test various hypothesis and gain better understanding of the biophysical properties of astrocytes and may be of interest to glial biologists, neurophysiologists, and computational biologists. However, the article are a few specific concerns detailed below.

We thank the Reviewer for the careful analysis of our manuscript and the constructive and encouraging comments.

Major

1. The outline and flow of the manuscript is logical but each section is lacking sufficient detail and it is difficult to ascertain what exactly was done and why.

Please see below.

2. The explanation of the model is lacking. The framework of the model is not well described, no equations are provided, the parameters used are not always provided, and their general stability has not been tested, i.e. how changing the measured or assumed parameters affect the simulations...

We appreciate that the main text had only a basic description of the model design. Perhaps, we did not make it clear that some detailed technical information had been provided in the 40-page ASTRO User Guide, which should have been supplied with the manuscript and is available online (<https://github.com/LeonidSavtchenko/Astro/>). Importantly, the vast majority of cell biophysical mechanisms involved have been well-tested and described across hundreds of pages of NEURON documentation (<https://www.neuron.yale.edu/neuron/docs>). However, we agree it is important to have such information available in the text or Supplementary materials, which we have now appended with some maths details of the key new mechanisms also adding relevant references/links.

We are slightly surprised by the question of parametric stability: the stability question is typical for analytical multi-stage kinetic models but is much less prevalent for multi-compartmental models employing direct finite-difference calculations and stepwise boundary-condition control. Firstly, NEURON generally rejects multi-compartmental geometries that are not computationally stable. Secondly, passive astroglial membrane mechanisms represent a near-linear dynamic system. Thirdly, when left alone or transiently challenged, these mechanisms bring the astroglial model system to a single equilibrated state, at the cell resting potential (i.e., a fixed point in the phase space) including stable K^+ and Ca^{2+} concentrations. Canonically, in such systems small parameter variations correspond to small variations in system's response. This is precisely what we observed throughout simulations (see below). One specific case of nonlinear dynamics was regenerative Ca^{2+} waves: in that case, however, we explicitly varied system parameters to understand variations in the outcome (new Fig. 6).

To address the Reviewer's comment more directly, we have carried out multiple additional tests to see how the physiologically plausible changes in various cellular parameters influence key biophysical traits of simulated cells (detailed below). These tests were summarised in a new manuscript section 'Variations in cell features: probing an impact on astroglial function' and illustrated by examples in the new Supplementary Figs. 6, 9-12.

E.g., how does changing the branch number within the estimation of data scatter (4-9 branches) affects the simulations?

Clearly, cell morphology shapes biophysical properties of the cell: branching structure is one of many such morphological influences. To predict how exactly the branch number affects astrocyte properties, one has to systematically compare multiple 3D-reconstructed cells: removing/adding branches computationally might not reflect reality faithfully enough.

To provide one example of the relationship between cell morphology and its functional properties, we compared two modelled astrocytes, one with the stem tree reconstructed from an *in situ* experiment (Fig. 1a-d), and the 'typical CA1 astrocyte', with the neurogliaform stem tree adjusted to match the average features of CA1 astrocytes (Fig. 1e). The two cells feature, respectively, 10/27 and 8/33 primary/secondary branches (short protrusions are recognised by computer as branches). In both cells, we populated branches with nanoscopic processes that have the same tissue volume fraction (VF) distributions, geometries, and biophysical properties (as illustrated in Figs 2-3). Thus, the two models had similar nanoscopic architecture but different stem tree branching patterns.

Next, we compared three basic biophysical features between the two cells: membrane voltage spread, input resistance, and Ca^{2+} wave generation. Simulations revealed subtle differences (new Supplementary Fig. 9), reflecting what one might call 'parametric stability' of the outcome.

How does changing the tissue volume fraction within the standard error (7-13% according to Figure 3b) affect the morphology of the cell and its biophysical properties, etc.

There must have been a misunderstanding. The experimental data (Fig. 3b, blue dots) indicate that the VF occupied by an astrocyte decreases from ~13% to ~7% *with the distance from the cell soma*. This is exactly what our 'average' cell model example reproduces (Fig. 3b, red dots): in this context, there is no standard error. Nonetheless, to address the Reviewer's request, we have simulated ~20% astrocyte 'swelling' (by homogeneously increasing the width of all cell processes; other parameters unchanged) and asked how this affects its basic properties. The outcome indicates some clear yet moderate consequences of swelling (new Supplementary Fig. 10). The data have been added to the manuscript.

3. It would be useful to use to test the model by testing a few predictions. For example:

a. How does the model predict the diffusion of larger molecules with lower diffusion coefficient such as dextran-conjugated alexa?

To illustrate the fact that slower diffusion means slower equilibration, we have simulated the scenario of astrocyte dye loading in whole-cell. The outcome provides a quantitative illustration of how reducing the effective obstacle-free diffusion coefficient from 0.6 to 0.05 $\mu\text{m}^2/\text{ms}$ (values typical of small ions and 2-3kDa molecules, such as Alexa 488-dextran, respectively) slows down dye equilibration (new Supplementary Fig. 11). In these tests, the contributing role of diffusion escape via gap junctions (for the smaller molecules) was not tested: this requires specific experiments to constrain the gap junction diffusion sink properties. Furthermore, diffusion of larger molecules is likely to be further slowed down by steric and viscous hindrance inside cells: again, matching simulated and experimentally imaged fluorescence profiles should determine the effective diffusion coefficient with great accuracy.

b. How does blocking potassium channel block affects the current of glutamate uncaging?

To reiterate our answers to Reviewer 1 above, we note that, because of the highly leaky astroglial membrane, local transporter current hotspot will stay local, with little effect on the voltage landscape (updated Fig. 4f). Therefore, adding $\text{K}_{\text{ir}}4.1$ should have little effect on the passive voltage landscape unless $\text{K}_{\text{ir}}4.1$ conductance exceeds substantially its experimental range. Having incorporated the full $\text{K}_{\text{ir}}4.1$ kinetic mechanism^{1,2} into the model, we have now carried out additional tests to illustrate this phenomenon (new Supplementary Fig. 6a). Relevant explanations have been added to the text, with the qualification that, although $\text{K}_{\text{ir}}4.1$ represent a major membrane mechanism in astroglia, additional experimental studies are required to assess the contributing role of other membrane currents, such as ion exchangers, pumps, and semi-channels.

c. How does change in astrocytic volume affect its biophysical properties?

As discussed above, we have simulated ~20% astrocyte 'swelling' (by homogeneously increasing the width of cell processes throughout the cell, no other changes) and asked how this affects its

basic physiological features. The outcome indicates some clear yet moderate consequences of swelling (new Supplementary Fig. 10).

d. How does block of addition of intracellular calcium buffer affects the propagation of calcium waves?

This is an important issue, which we had partly explored in the context of simulating Ca^{2+} waves seen *in vivo* (Fig. 6c). To address this more explicitly, we have carried out additional tests simulating Ca^{2+} wave generation with and without a mobile Ca^{2+} buffer added to an endogenous stationary buffer (new Fig. 6a; Supplementary Movie 4). Intriguingly, adding only a modest amount of the mobile buffer significantly slowed down the wave propagation (Fig. 6a), also increasing the time period of $[\text{Ca}^{2+}]$ elevation. An accurate interpretation of these interesting results clearly requires a separate, dedicated study.

e. Etc.

Indeed, there are potentially countless scientific questions that one could explore using ASTRO, with or without incorporating their own experimental constraints and observations. Our aim was precisely to provide such an investigative tool while showing several important examples of its application.

4. The model is based on an isolated astrocyte. This is, of course, is not the case for astrocytes in the hippocampus or most other brain regions which show a high degree of coupling to one another. Indeed, the coupling of astrocytes likely influences most of the biophysical properties this program attempts to address. This parameter is not really considered in the new model.

The gap-junction feature, with diffusional and electric leak settings, had previously been explained and illustrated in the ASTRO User Guide (p 27) supplied with the manuscript, with the working menu options available in the program. We have now added separate manuscript section explaining this feature.

In addition, to obtain some basic experimental constraints in this context, we have carried out patch-clamp experiments in CA1 astroglia in which gap junctions (and probably hemi-channels) were blocked by carbenoxolon (CBX, 50 μM). This increased cell input resistance by ~30% (new Supplementary Fig. 8), consistent with the previous measurements³. However, throughout the study simulations were carried out with the gap-junction conductance contributing to the overall cell membrane conductance by default, to reflect a common physiological scenario. Explanations have been added to the text.

Specific comments

1. The title of the manuscript is "Disentangling astroglial physiology with a realistic cell model". However, the model described mainly deals with the biophysical properties of the astrocyte, and not with its physiological function (i.e. extracellular potassium and neurotransmitter buffering, blood-brain barrier maintenance, metabolic support of neurons etc.). A title such as "Disentangling astroglial biophysical properties with a realistic cell model" is more suitable.

We generally agree and have changed the title into "Biophysical underpinning of astroglial physiology probed with realistic cell models".

2. The introduction lacks any summary of previous attempts to model astrocytes.

We have added a brief summary of the previous attempts to model astrocyte function, in various contexts.

3. The "Reconstructing astrocyte gross morphology" section in the Results is missing important information which makes the readers interpretation difficult. i.e Why were several programs were used for the 3D reconstruction? Was one better than the others? Did they converge? What

parameters were extracted and used in the model? Which of the approaches were used for stem tree construction? This section needs to be re-written more clearly.

There are several tools available in the public domain to provide 3D reconstruction of the cell 'stem tree' from 3D stacks of fluorescent images. Given the same imaging settings and the same user-defined criteria for branch identification, such programs normally provide reconstructions that are indistinguishable. We have now clarified that we used ImageJ Fiji for 3D tree reconstruction from raw image z-stacks and then converted the reconstructed morphology into NEURON format using Vaa3D (Allen Institute). Thus, we were able to use freeware throughout. Other investigators might prefer commercially available Neurolucida, which has been a tool of choice for many NEURON users.

Likewise, there are several programs that provide 3D reconstruction, including surface rendering, from stacks of ultrathin serial EM sections. Again, the outcome depends on the user's settings and criteria rather than on the program type per se.

These 3D reconstruction programs replicate the *entire geometry* of the reconstructed cell branches, normally by generating a dense meshwork of 3D coordinates on the cell membrane surface points. It is up to the user to extract particular morphological parameters from such massive data sets, be it branch numbers, diameters, lengths, surface areas, surface-to-volume ratios, etc. We have expanded on these issues in the text.

4. More detail is needed to understand how the 'transitional' slab cross section was calculated.

We have added a new explanatory figure panel (new Fig. 2d) and further details to the text and Fig. 2 legend.

5. The authors should more clearly define the difference between stem branches and nanoscopic protrusions. It is not clear how the nanoscopic processes were added to the stem tree of the astrocytes, and how it was later adjusted to fit the calculation of volume fraction and surface to volume ration.

The 'stem tree', featuring stem branches, is the cell arbour structure that could be seen and identified in the light (fluorescence) microscope, above the diffraction limit (i.e. branches thicker than 0.3-0.4 μm). The rest of morphology is represented by multiple nanoscopic protrusions that are blurred in the light microscope; however, their shapes and tissue-filling properties can be described by the statistics acquired using 3D EM and other methods, as detailed in the manuscript. Based on these experimental statistics, ASTRO generates nanoscopic protrusions stochastically and 'plants' them on stem branches in accord with the user-defined density and size. These latter two parameters, provided in the ASTRO (scripted SeedNumber and menu Leaf Number, respectively), enable one to adjust the cell nano-morphology to fit the experimental distribution of volume fraction (VF) across different distances from the cell soma (simulated VF distribution is displayed in during the shape-generation process). We have revised the text (pp. 8-9) to clarify these issues.

In addition, as above, it is not clear whether these parameters are stable, or a slightly different parameters could lead to a significant change in the morphology of the model astrocyte.

As noted above, we are not entirely sure why 'parametric stability' causes concern. The answer to the specific question above could be found throughout the manuscript: ASTRO generates nano-morphology stochastically, based on the statistics provided. Thus, in many cases, simulation tests involve slightly varying cell morphology (Fig. 4c provides a direct example). Furthermore, we have carried out multiple additional tests to see how the physiologically plausible changes in various cell parameters influence key biophysical traits of simulated cells, and found no 'runaway' phenomena (e.g., Supplementary Figs. 6, 9-12).

6. In Figure 3, what is the diffusion coefficient used for the modelling of FRAP? Also, the FRAP did not seem to recover much, if at all. Perhaps longer recovery times are needed. In Figure 3g, it is not clear what is normalized FRAP time. Is it the τ of recovery?

The diffusion coefficient value of $0.3 \mu\text{m}^2/\text{ms}$ has been added to the text. The approximate normalised FRAP time constant in these tests was $0.1 \mu\text{M}/\text{ms}$ but our aim was to fit the entire time course of bleaching-recovery as observed: the value itself is only a scaling factor.

Also, the stability of the rate of Alexa line-scan photobleaching is not tested for stability.

There must have been a misunderstanding. The experimental data in the original Supplementary Fig. 4 (new Supplementary Fig. 5) show that full fluorescence recovery occurs within <60 s from the FRAP onset. Furthermore, 60 s after the first cycle, the second FRAP cycle fully reproduces the first cycle kinetics of photobleaching and recovery (Supplementary Fig. 5c). These data provide evidence for full recovery and FRAP stability in the current settings. The text has been revised to clarify this.

7. The "Passive electrical properties of astrocytes" (Figure 4a-c) is poorly explained, especially how the G_m parameter estimate was obtained.

First, we estimated G_m value experimentally, in the classical patch-clamp test using outside-out patches of the cell plasma membrane (Fig. 4b). Second, we asked if our astrocyte model constrained by two *other independent* experimental parameters, input resistance R_i and total membrane area S_{mem} , can reproduce this value of G_m , in accord with $G_m = (S_{\text{mem}}R_i)^{-1}$. Because modelled nanoscopic processes are generated stochastically, we have obtained a small representative sample of the astrocyte in question: reassuringly, this sample showed variation in the cell volume (Fig. 4c) similar to that reported in the literature (see text). In the sample, the mean value of G_m calculated as $(S_{\text{mem}}R_i)^{-1}$ (Fig. 4c, dotted line) was indistinguishable from the directly measured experimental G_m (Fig. 4b). The test thus confirms that our model reproduces 'correct' experimental value of G_m based on other independent constraints. This suggests the realistic nature of the simulated nanoscopic architecture. We have expanded the text to clarify this.

8. In the section "Membrane voltage landscape generated by local glutamate uptake", it is not clear how GLT1 were "scattered". Also, specific GLT1 kinetics used here should be noted.

GLT1 were scattered uniformly on the cell surface. This and the specific reference to the GLT1 kinetics have been added to the text.

9. In the section "Potassium uptake and redistribution inside astrocytes", the kinetics and parameters of the first order pump used for mimicking potassium intrusion should be noted in text, as well an explanation in text of the parameter J_{ext} . Units of J_{ext} should be noted in Figure.

We have now incorporated a full-scale $K_{\text{ir}}4.1$ mechanism into ASTRO-NEURON. Based on that, we have carried out new simulations illustrating the relationship between extracellular K^+ rises, membrane voltage dynamics, channel function and intracellular K^+ redistribution, in a physiologically plausible scenario (new Fig. 5). The old Fig. 5 data have been updated and moved to the new Supplementary Fig. 7 as an additional illustration (here, J_{ext} was replaced with pump parameter K_p for clarity).

10. The modelling uses in vitro parameters of CA1 astrocytes. However, modeling of calcium waves uses in vivo measurements of somatosensory cortical astrocytes. A more appropriate way to do this is to measure calcium waves in ex vivo CA1 astrocytes.

There is an ongoing debate in the field, with some groups suggesting that Ca^{2+} waves seen in *ex vivo* settings are artefactual. Because it was not technically feasible to image intracellular Ca^{2+} waves in the hippocampus *in vivo* without significant mechanical invasion of the brain (with poorly controlled artefacts), we imaged somatosensory cortex astroglia, which appear to show remarkable similarities in their major morphological features and territorial volumes, across ages, with hippocampal astrocytes⁴. We have added explanations to the text.

11. The authors should add to their discussion several topics: what is this model good for, and what are its weaknesses? How one can extract physiological effects from this biophysical model?

We have added these points to the discussion. In brief, in our study we tried to show examples where the biophysical model could evaluate how fast molecular diffusion exchange occurs within the astrocyte, whether and how local electric events can control voltage-sensitive membrane mechanisms across the astrocyte, how fast K^+ could enter the cell and how fast it could equilibrate inside it, what key parameters control Ca^{2+} waves, how one should interpret Ca^{2+} imaging recordings, etc. In general, biophysical cell models are there for investigators (a) to test whether their interpretation or hypotheses are biophysically plausible, (b) to understand microscopic spatiotemporal profiles of ion currents and molecular fluxes which cannot be registered experimentally, (c) to predict the relationships between cellular features (morphology, Ca^{2+} buffering, channel current density, molecular transport, etc.) and physiological events registered experimentally.

How can this model be further developed (e.g. addition of gap junctions, end feet, changes in volume...

Both the gap junctions and endfoot features were present in the original ASTRO User Guide and the program (but not in the main text): they have now been added as separate sections to the manuscript. The volume change feature has been explained and illustrated in the manuscript.

.. changes in synaptic coverage? How is the tripartite synapse is represented in the model?

ASTRO interface has direct access to the full library of synaptic and non-synaptic receptor mechanisms enabled by NEURON. Therefore, arbitrary patterns of 'synaptic coverage' could be modelled using standard NEURON Menus. In addition, excitatory synaptic transmission could be mimicked using the glutamate uncaging menu, as fully explained in the text and Supplementary ASTRO User Guide. Clearly, the investigator is supposed to incorporate a specific, experimentally constrained kinetic mechanism for the receptor action of interest. Judging by the numerous examples from neuronal physiology, each such quest will require a separate, dedicated experimental study to establish key constraints. We have expanded the Discussion accordingly. We note that modelling blood vessels, synapses, microglia, etc. was outside the scope of the present study. However, they will be included in the future versions of ASTRO.

Minor comments:

1. In many figures it would be easier to understand if there was an accompanying supplementary movie of the modelling, and not just a snapshot: e.g. Figure 2e, Figure 7

We have added eight movies, as suggested, to illustrate various experimental observations and simulation tests (Supplementary Movies 1-8).

2. What is the red box in Figure 1e?

We have removed this NEURON interface glitch.

3. Figure 2b has bad colors, especially yellow on white background.

We have added a darker background to make it clearer.

4. How thick are the slices in Figure 2c?

They follow the actual 60 nm thick serial EM sections. Figure legend expanded accordingly.

5. What are the different shades of blue in Figure 2d?

These shades depict the facets of stacked polygonal slabs seen from one side. The new Fig. 2d illustration has been added for clarity.

6. In Figure 2f, at which plane where the particles release, and where was the diffusion flux measured? Also, the use of ampere instead of $e\text{ ms}^{-1}$ (use electrons instead of generic charged particle) is easier to comprehend. Add electrical parameters to Figure legend as in Supplementary Figure 3.

It was explained in Fig. 2g legend that particles were injected at the bottom, and their arrival time at the top is registered. In Monte Carlo electrodiffusion simulations, Amps are not very useful to illustrate particle numbers and speeds. Further details added as requested.

7. Where is representative example in Figure 3f measured?

A protoplasmic astrocyte in CA1 area, *stratum radiatum*. Details added.

8. Figure 4 Y axis title should be "Input" not "Inpur".

Corrected.

9. In Figure 6b, scale is not provided, especially important in the lower panel ("dC/dt").

False colour scales added.

10. In page 8 there are references to Supplementary Information. However, only Supplementary Figures were provided.

We have provided an illustrated ASTRO User Guide as Supplementary information for the manuscript. We assume this had been supplied to the Reviewers in the previous round.

11. Page 11 second paragraph 5th sentence, should be "Kir4.1" nor "Kir1.4".

Corrected.

12. In Methods, electrophysiology ex vivo section, last paragraph, 7th sentence there are two repetitive words: "Ascorbic ascorbic..." and "Sodium sodium...".

Corrected.

13. In Methods, Fast fixation and DAB staining of recorded astrocytes, 4th sentence, it is not clear what "infiltrated" means".

Replaced with 'submerged' or 'embedded'.

Reviewer #3 (Remarks to the Author)

We thank the Reviewer for the careful consideration of our manuscript.

This study uses the NEURON computational environment to model basic properties of astrocytes. The report is well written and easy to read, but I have several concerns. The premise for the model is that several basic properties of astrocytes are neglected.

There must have been a misunderstanding. Our model builder replicates *known* morphological and biophysical features of astroglia, precisely to enable exploration and testing of the *yet unknown* features.

Also, the modeling does not provide interesting questions, but strictly replicate the biology already reported. A specialized glia journal may be a better fit for this study.

The Reviewer's comment 'the model strictly replicates the biology already reported' seems to directly contradict the preceding comment that 'several basic properties of astrocytes are neglected'.

One of the reasons to develop the tool was the growing need to validate what seem to be biophysically implausible interpretations that populate published observations on astroglia. We are convinced that introducing a biophysical testing platform to astrocyte physiology must play a similar role to that played by the neuronal modelling tools in modern neurophysiology.

Major critique:

(1) Astrocytes are expressing a large number of gap junction that couple the fine astrocytic

processes belonging to the same cell and to its neighbors together. Gap junctions are fully permeable to Ca^{2+} , IP_3 and small tracers. This fundamental feature of astrocytes that will affect Ca^{2+} signaling, K^+ buffering, glutamate uptake and more is neglected

The gap-junction feature, with diffusional and electric leak parameters, had been explained and illustrated in the ASTRO User Guide, which was supplied with the manuscript. We have now added to the manuscript a separate gap junction section with relevant details.

(2) Oscillatory or wavelike propagation of Ca^{2+} waves within single astrocytes is a sign of pathology. It is often observed in slices or in anesthetized mice after poor surgery or in response to excessive light stimulation. Awake behaving mice display synchronous increases in Ca^{2+} that is blocked by α_1 receptor antagonists. Why model data based on non-optimal preparation when this study is presented as a model of normal physiological behavior?

We are slightly baffled by this comment. We are familiar with the recent work from Maiken Nedergaard group and have cited it. Nonetheless, to answer the comment, we have added to the manuscript a movie recorded in the cortex of an awake animal on a treadmill, in which one could detect Ca^{2+} waves within individual astroglia (using lckGCaMP6f): these waves intermittently occur between large, field-wide synchronous Ca^{2+} elevations apparently associated with motion (Supplementary Movie 6). Single-cell waves do appear less prominent and less frequent than those in the anesthetised animal yet they seem to show similar Ca^{2+} kinetics. As mentioned in the text, it is precisely to avoid contamination with large synchronous events that we focused on Ca^{2+} wave recordings in anesthetised animals. Importantly, the purpose of our exercise was to establish the Ca^{2+} buffering capacity of astroglia, which is highly unlikely to be affected by anaesthesia. Explanations have been added.

(3) Astrocytes are highly polarized cells extending 1-3 vascular endfeet. These processes are a key to astrocytic function, but not included in the model. Why not?

Again, the endfoot feature had previously been explained and illustrated in the ASTRO User Guide (but not the main text), with the fully working menu options available in the program. We have now added to the manuscript a separate section, with relevant details pertinent to the endfoot feature. We note, however, that a systematic exploration of the endfoot requires a separate dedicated study.

(4) The model data are not providing any questions or prediction that would be interesting to test.

We are struggling to understand the basis for such a statement. Our selected modelling tests have provided first quantitative insights into the voltage and current spread across the astrocyte membrane, cell Ca^{2+} buffering capacity, relationships between extracellular K^+ rises and intracellular K^+ redistribution, the rate of intracellular molecular diffusion in 3D, glimpses of the free Ca^{2+} kinetics (which is drastically different from the kinetics of fluorescent readout using Ca^{2+} -sensitive indicators), etc. ASTRO is currently the only tool that could, in theory, decipher the dynamic, space-time relationships among ion currents and molecular fluxes inside astroglia that result from activation of various channels and pumps. Whilst all this needs to be constrained by experimental data, theoretical modelling is essential for understanding the mechanics of microscopic and nanoscopic interactions inside cells.

To reiterate, we are convinced that introducing a biophysical testing platform to astroglial physiology should play a similar role to that played by neuron-modelling tools in modern neurophysiology.

Reviewer 4 (rebuttal added at short notice)

We thank the Reviewer for the careful consideration of our work and insightful comments

The main achievement of the manuscript is developing a simulation tool for modeling calcium and potassium dynamics in a detailed astrocyte morphology of astrocytes. This software platform is shown to interphase well with 'Neuron', a popular neuron modeling tool that is used to simulate neuronal activity in realistic morphologies.

As far as I know, there is no extant tool box available that allows for modeling astrocytic calcium and potassium dynamics in realistic spatial domains. As a add-on functionality to 'Neuron', it will serve a large user base. Decoding fluorescent calcium dye measurements to calculate real calcium signal is an important contribution of this paradigm.

While we fully agree with the Reviewer, we also note that the astroglial literature is not devoid of biophysically questionable claims pertinent to $[K^+]$ redistribution, membrane voltage/current spread, remote electrogenic effects of glutamate transport, etc. We do hope that our modelling approach will help to provide a biophysical basis for resolving such issues.

Substantial effort and time in the manuscript is spent on creating a canonical geometrical representations of an astrocyte that can be morphed into 'Neuron'. This required that 1) statistical description of the cellular microphysiology can be justified and modified to suit the user 2) ensure the cylinder-compartment discretization of space as required by 'Neuron' is a reasonable assumption that does not compromise accuracy. Here a exhaustive list of key morphological features and bounds on physiological parameters should have been made available.

We did not fully understand the latter request. Unlike the vast majority of astrocyte modelling studies, we actually *measured* morphological and functional features of the cells under study trying to constrain some key model parameters. Perhaps we did not make it explicitly clear, but the model construction does indeed run through an 'exhaustive list' of key features, in a fairly systematic way: from 3D-reconstructed stem-tree (main-branch) morphology, to nanoscopic process statistics obtained using 3D EM, to volume fraction distribution obtained using 2PE microscopy, to input resistance and unit membrane resistance obtained using patch-clamp, to glutamate transporter current simulations that match glutamate 2P-uncaging experiments, to Ca^{2+} buffering and Ca^{2+} wave simulations that match our experimental recordings, *in situ* and *in vivo*. We are struggling to find any astroglial study that would feature a similar degree of the experiment-versus-modelling appraisal. To follow the Reviewer's request we, have added a Table summarising key morphological features pertinent to the astroglia under study (Supplementary Table 1).

Also, the "MATLAB based algorithms" that are used in 'ASTRO' to generate nanoscopic processes is of potential scientific value, however not discussed at all.

Transforming 3D-reconstructed astroglial processes into NEURON-compatible processes, with a Monte Carlo test for biophysical compatibility, is indeed a scientifically important step. The underlying algorithm is, however, relatively straightforward: we have expanded the text adding basic formulas and a further explanatory illustration (Fig. 2). The Monte Carlo routines for electrodiffusion tests in arbitrary geometries have been detailed and discussed in our previous work⁵⁻⁷: these references have been added. In the end of morphological sampling, MATLAB generates a file containing the stats for the radii of cylindrical compartments constituting nanoscopic processes ('leaves' and 'stalks'). The output file NanoDistrFromFile.hoc is then used by NEURON to generate nanoscopic processes that fill in the 3D space (based on random number generator), as detailed in the ASTRO User Guide (User Guide Fig. 19 and related information). We have clarified this in the text, removing the emphasis on MATLAB where required.

Unfortunately, once the model is ready, its falls short of any serious investigation for new insights. The authors carry out a few illustrative simulations in the model that describe intracellular potassium and calcium dynamics, but no interpretation or consolidated view on its significant is offered. Very little time is spend in exploring the various components of the model, model parameters are not studied systematically (except for buffering concentration and K_D).

There must have been a misunderstanding. Realistic 3D cell morphology and multi-compartmentalisation of space in the model opens up a completely new horizon of exploration, which has not hitherto been available. At the same time, complex geometry means the parameter-space dimensions are expanded considerably. Therefore, any serious hypothesis testing requires some considerable effort to constrain free model parameters by experimental data. We note that it took us years to set up experimental studies combining 3D EM, patch-clamp electrophysiology, 2PE imaging and uncaging, viral transduction of Ca^{2+} indicators, etc., *in situ* and *in vivo*, to be able to come up with at least some critical constrains of the model. With all such constrains, we were able to show basic properties of voltage/current spread in the astroglial membrane (a highly contentious issue), how local glutamate transport can affect membrane potential (a highly contentious issue), the effective speed of intracellular diffusion and its interaction with the extrusion mechanism, astroglial Ca^{2+} buffering properties, and how the recorded fluorescence Ca^{2+} signals can represent the underlying Ca^{2+} activity (a highly contentious issue). Nonetheless, in accord with the Reviewer's suggestions, in the revised version we have carried out a number of additional simulation tests exploring the effects of astroglial $\text{K}_i4.1$ K^+ channels with realistic kinetics (new Fig. 5, new Sup Fig. 6), heterogeneous (immobile + mobile) Ca^{2+} buffers (new Fig. 6a), cell morphology variations (Sup Fig. 9), and cell volume changes (Sup Fig. 10) on the basic physiological features of astroglia.

A objective of a good realistic model allows you investigate and predict behavior under different stimuli and protocols and extend the understanding of experimental findings. This is precisely where the manuscript is lacking. New insights that could be obtained using this modeling paradigm and its functional implications to neuronal activity are not investigated at all.

Again, we are not sure we fully understand this comment: our work present multiple tests in which experimental observations match model predictions or are used to constrain key model parameters: Fig. 1e (branch morphology), Fig. 3b (volume fraction distribution), Fig. 3g (FRAP model vs experiment), Fig. 4b-c (unit membrane conductance, model vs experiment), Fig. 4e (glutamate uncaging/transport, model vs experiment), Fig. 6b-e (Ca^{2+} waves, model vs *in vivo* experiment), Fig. 7 (Ca^{2+} signals, model vs acute slice data). Again, we hope the Reviewer appreciates the extent of our effort aimed at the comparison between our modelling and our own experimental observations in astroglia.

Whereas experimental methods used to explore morphology have been detailed well and justified, the theoretical methods are not outlined. Unfortunately, it ends up being a software package wherein the algorithms used remain a black box.

We do appreciate that the main text had only a basic description of the model design. Perhaps, we did not make it clear that some detailed technical information had been provided in the 40-page ASTRO User Guide, which should have been supplied with the manuscript and is available online (<https://github.com/LeonidSavtchenko/Astro/>). Importantly, the vast majority of cell biophysical mechanisms involved have been well-tested and described across hundreds of pages of NEURON documentation (<https://www.neuron.yale.edu/neuron/docs>). However, we agree it is important to have such information available in the text or Supplementary materials, which we have now appended with some maths details of the key new mechanisms also adding relevant references/links to the more technical info sources. We will be happy to expand this further if this is deemed of practical use to the reader.

[...] Here is list of more specific critique

1) Abstract itself does not reflect any new scientific insight but indicates that the tool box is the central achievement. For example “evaluate some of the basic features of astroglial physiology inaccessible in experiments such as membrane voltage propagation, K^+ redistribution dynamics, or intracellular Ca^{2+} buffering capacity”, What is it that the evaluation leads to is not mentioned and its effect on function is not investigated. This is a crucial missing component throughout

given that this is a modeling study. The abstract goes on to say “paves the way for mechanistic interpretations ...” However the nature of these interpretations is not elaborated upon at all.

The word limit provides little room for elaboration in the abstract, which normally aims at a general readership. We have however modified it, to be less verbose about the 'toolbox' and more specific about the novel insights the manuscript provides.

2) With all morphological measurements being carried out here, a quantification of a canonical astrocytic morphology with relevant statistical variation in a table should be shown.

We have added a brief table of morphological parameters, as requested (Supplementary Table 1). Note, however, that some of the key morphological features require distribution-like representation rather than mean values and moments, and have been presented accordingly (Fig. 1e, Fig. 3b, Sup Fig. 2-3).

3) Page 3, “We have therefore developed (MATLAB-based) computational algorithms and software that, firstly, use experimental data to recreate the space-filling architecture of nanoscopic astroglial processes, and, secondly, make this cell architecture NEURON-compatible.” These algorithms are not described.

As mentioned above, we have expanded the text re the simple algorithm of nano-process transformation from 3D-reconstructed to NEURON compatible shapes, adding further explanatory illustration (Fig. 2) and basic formulas. The Monte Carlo routines for electrodiffusion tests in arbitrary geometries have been detailed and discussed in our previous work⁵⁻⁷: these references have been added. In the end of morphological sampling, MATLAB generates a file containing the stats for the radii of cylindrical compartments constituting nanoscopic processes ('leaves' and 'stalks'). The output file NanoDistrFromFile.hoc is then used by NEURON to generate nanoscopic processes that fill in the 3D space (based on random number generator), as detailed in the ASTRO User Guide (User Guide Fig. 19 and related information). We have clarified this in the text, removing the emphasis on MATLAB where required.

4) Page 4, “which we have equipped with multiple additional functionalities specific to astroglia.” What additional functionalities specific to Neuroglia is not mentioned. Especially, the development of the toolbox is the main achievement here.

There must have been a minor misunderstanding. The manuscript and the Supplementary ASTRO User Guide explains in great detail what astroglia-specific, extensive algorithms and menus have been added to NEURON: (a) planting, sizing and volume-filling by nanoscopic astroglial processes; (b) FRAP menu; (c) endfoot menu; (d) gap junction menu; (e) extracellular K⁺ menu; (f) glutamate spot-uncaging menu; (g) Ca²⁺ signal generation and 'fluorescence detection' menu, among others. We have summarised this in the text as suggested.

5) Page 10, “Equipped with these experimental constraints, we simulated a simple voltage-clamp experiment, which predicted that a voltage signal generated at the soma should spread with an electrical 3D-space constant of 30-60 μm depending on the signal frequency (Fig. 4d, left). Intriguingly, a membrane voltage signal generated at the periphery was attenuated with a shorter space constant of 25-30 μm (Fig. 4d, right), the asymmetry previously reported in nerve cells”. Characterization of this 'intriguing' difference not investigated.

This asymmetry is a straightforward consequence of the current sink distribution near the signal location. Its 'intriguing' feature is not in the mechanism but in the potentially different membrane effects of similar current signals generated at different location on the cell tree. We have added an explanation to the text.

7) Page 10, “As the most likely scenario of astroglial depolarisation relates to the accumulation of extracellular K⁺, these data should help to understand the electrogenic membrane mechanisms

involved.” The statement is too abstract. The mechanisms involved should be quantified using the developed model.

Our revised version contains new mechanisms enabling extracellular K^+ rises and full Kir4.1 kinetics. We have significantly expanded our tests and explanations, exploring the scenarios of K^+ buffering and glutamate uptake (new Fig. 5 and Sup Fig. 6).

8) There are more astrocyte specific models for IP₃ and RyR with very different behavior. However off the shelf models from “Neuron” are used. This may be a problem for classification of calcium dynamics. A justification of the models used for receptors should be given as they can crucially change the results.

There have been a large variety of (single-compartment) kinetic astroglial models involving IP₃-receptor (or RyR) - dependent Ca^{2+} signalling in astroglia. Most of these models ignore structural complexity of astroglia, spatial heterogeneity of Ca^{2+} sources, intra- or extracellular concentration gradients, or real imaging data obtained with high-affinity Ca^{2+} indicators in the presence of endogenous Ca^{2+} buffers. It is therefore difficult to rely entirely on the receptor kinetic estimates based on such simplifications. However, to follow the Reviewer's request we have incorporated two established, astroglia-related IP₃ receptor-signalling models into the optional ASTRO-NEURON mechanisms, one from De Pitta et al.⁸ (which includes glutamate signal dependence), and one from Fink et al.⁹. The instructions on how to switch between the three IP₃ signalling options (original, DePitta's, and Fink's) are briefly explained in the text and outlined in more detail in the Supplementary ASTRO User Guide (p. 30).

9) Page 11, “Clearly, precise parameters of K^+ entry and extrusion will depend on the experimental circumstances that determine the kinetics of extracellular $[K^+]$, K^+ channel activity, and the voltage landscape of astroglial membrane 16. Once these and related quantities have been empirically constrained, ASTRO should provide important clues to the spatiotemporal dynamics of the intra-astroglial K^+ buffering and redistribution, on the scale from individual synapses to hundreds of microns.” These precisely are the kind of computational experiments that can be carried out here to acquire a more complete understanding. A wide range of parameters for K^+ entry, extrusion and voltage landscape can be easily studied that would potentially provide a quantitative understanding complete range of potassium dynamics.

We generally agree: the revised version contains new mechanisms enabling extracellular K^+ rises and full Kir4.1 kinetics. We have significantly expanded our tests and explanations, exploring the scenarios of K^+ buffering and glutamate uptake (new Fig. 5 and Sup Fig. 6).

11) A systematic calculation of changes in calcium signaling with different clustering and inter-cluster distance of IP₃ receptors and RyR using the model should be done.

To follow this specific request, we have simulated several scenarios in which IP₃-dependent receptor activity was clustered (spaced at 1, 2, or 4 μm apart), with the average IP₃ level and all other model parameters kept unchanged. These data have been added as Supplementary Fig. 12: they suggest that clustering the IP₃ activity facilitates the triggering of local $[Ca^{2+}]$ rises, which are strongly amplified in Ca^{2+} sensitive fluorescence signals (Supplementary Fig. 12b). The latter highlights the highly non-linear relationship between $[Ca^{2+}]$ dynamics and registered fluorescence signals. Clearly, further exploration of these phenomena requires additional experimental constraints, such as patterned spot-uncaging of IP₃ inside astroglial processes (currently underway).

13) Effects of homogenous and heterogenous buffering should be classified

There are no reliable experimental data shedding light on the heterogeneity of Ca^{2+} buffers in astroglia: therefore, a classification attempt would represent a highly unconstrained theoretical exercise engaging free model parameters. However, it is an important question, and to test some basic effects, we have carried out additional simulations in which the influence a mobile Ca^{2+} buffer on Ca^{2+} wave propagation has been explored (new Fig. 6a).

References

1. Seifert, G., *et al.* Analysis of astroglial K⁺ channel expression in the developing hippocampus reveals a predominant role of the Kir4.1 subunit. *J Neurosci* **29**, 7474-7488 (2009).
2. Sibille, J., Dao Duc, K., Holcman, D. & Rouach, N. The neuroglial potassium cycle during neurotransmission: role of Kir4.1 channels. *PLoS Comput Biol* **11**, e1004137 (2015).
3. Wallraff, A., *et al.* The impact of astrocytic gap junctional coupling on potassium buffering in the hippocampus. *J Neurosci* **26**, 5438-5447 (2006).
4. Grosche, A., *et al.* Versatile and simple approach to determine astrocyte territories in mouse neocortex and hippocampus. *PLoS One* **8**, e69143 (2013).
5. Savtchenko, L.P. & Rusakov, D.A. The optimal height of the synaptic cleft. *Proc Natl Acad Sci U S A* **104**, 1823-1828 (2007).
6. Savtchenko, L.P., Sylantsev, S. & Rusakov, D.A. Central synapses release a resource-efficient amount of glutamate. *Nat Neurosci* **16**, 10-12 (2013).
7. Sylantsev, S., *et al.* Electric fields due to synaptic currents sharpen excitatory transmission. *Science* **319**, 1845-1849 (2008).
8. De Pitta, M., Goldberg, M., Volman, V., Berry, H. & Ben-Jacob, E. Glutamate regulation of calcium and IP3 oscillating and pulsating dynamics in astrocytes. *J Biol Phys* **35**, 383-411 (2009).
9. Fink, C.C., *et al.* An image-based model of calcium waves in differentiated neuroblastoma cells. *Biophys J* **79**, 163-183 (2000).

REVIEWERS' COMMENTS:

Reviewer #1 (Remarks to the Author):

This is an excellent revision that clarifies many issues pertaining to the model that were brought up by the reviewers. There are many unknowns about the properties of the fine processes of astrocytes and yet most of the impact of ion homeostasis by astrocytes is due to the properties of these fine processes. In fact, recent advances in imaging and in GECIs now allow glial researchers to see a surprising degree of calcium microdomain signaling in these distal processes. This model will provide an important and extremely useful framework for testing hypotheses concerning the interplay between calcium signaling, transporters, ion channel distribution and electrophysiology in the fine processes. I encourage publication of this manuscript as I think it will become a critical reference aid for visualizing current concepts and for testing the theoretical impact of cellular alterations. I personally look forward to having this tool at hand for exploring hypotheses about astrocytes.

Reviewer #2 (Remarks to the Author):

The re-submitted version of Savtchenko et al. is significantly improved. The additional text adds clarity and negates the need for those reading the manuscript to consult the supplemental user's manual. The scientific power of the ASTRO simulator is demonstrated in several new simulations (Supplementary Figures 6,9,10,11,12 and Figure 5). The Supplementary Videos add clarity of the paper and give specific examples of how the program may be used. The Discussion has been expanded considerably.

Minor comments:

In response to reviewer 1 the author states 'astrocytes are leaky because of the large membrane area', while astrocytes do have a large membrane area, they are also leaky because they express several leak K⁺ channels as well as Kir4.1 which all have high open probabilities at the resting membrane potential. Additionally, they are coupled to other astrocytes.

Figure 5 – Was the 0.1 mS/cm² of Kir4.1 the only conductance, or was it done on the background of the non-specific 1 mS/cm² leak current?

Page 3, line 42 – Should be 'Three' not 'Tri'.

Page 10 line 269 - "...and the modelled nano-shapes (Fig. 2f..." – seems like a reference to the wrong figure.

Page 12 line 348 semi-channels should be changed to hemi channels

Page 13 line 381 – Voltage-dependent should be erased, as Kir4.1 are not.

Page 14 line 404 – 'Fig. 5b', not 'Fig 4b'.

Page 14 bottom paragraph (and Supplementary Figure 7) – why was the constant rate of the potassium extrusion set at $K_p=0.5$ mM/ms? Is there a previous reference for that?

Page 33 line 1034, Figure 1 legend - at the end should be ' not , , .

Page 35 line 1124, Figure 4f legend – the left diagram represent current density, not current as in text.

Page 43 line 1438 – change 'Supplementary Figure xx' to 'Supplementary Figure 7'.

Figure 3b – measurement unit (s) should be annotated on the Y axis.

Figure 4d – color code of different frequencies does not seem to match the colors of the lines. Also, it also might be better to start the X-axis ($x=0$) of the right panel at the signal initiation point, so space constant could be more intuitively extracted.

Figure 6c lower panels – the colors in the scale bar don't seem to correlate to the actual colors.

Figure 6d – units ($\mu\text{m/s}$) should be noted for the Y-axis. Also, does the simulated data fit the in vivo measurements?

Figure 7f – The 2+ in the $[\text{Ca}^{2+}]$ symbol in the figure is truncated.

Supplementary Figure 2 – color code should appear in the Figure.

Supplementary Figure 6d – There is a representation of only one case, despite color code for all three.

Supplementary Figure 6d legend, should be 'Three' not 'Thee'.

Supplementary Figure 9b - units should be noted for the Y-axis

Supplementary Table 1 – Do the values after the \pm sign represent standard deviation or standard error?

Reviewer #3 (Remarks to the Author):

The authors have powerfully addressed my critique and I have no further comments. I personally do not think that modeling based on the biophysical properties of astrocytes will add much but the future may show differently. These tools have been important in the understanding of neuronal properties

Reviewer #4 (Remarks to the Author):

The authors have addressed most of my concerns. I am glad to see new simulations and results wherein the model has been put to good use to get novel insights on potassium signaling, calcium buffering and glutamate uptake in the revised manuscript. The effect of changing morphology has also been explored. As recommended the explanations are more detailed and theoretical methods appear in the main manuscript. I would also suggest that in the abstract, instead of somewhat vague, "...some basic relationships between free Ca^{2+} dynamics and experimental readout of fluorescent Ca indicators", a more specific key relationship be stated here.

Point-by-point response to Reviewers

Reviewer #2 (Remarks to the Author):

Minor comments:

In response to reviewer 1 the author states 'astrocytes are leaky because of the large membrane area', while astrocytes do have a large membrane area, they are also leaky because they express several leak K⁺ channels as well as Kir4.1 which all have high open probabilities at the resting membrane potential. Additionally, they are coupled to other astrocytes.

Clarified in p. 10 3rd para.

Figure 5 – Was the 0.1 mS/cm² of Kir4.1 the only conductance, or was it done on the background of the non-specific 1 mS/cm² leak current?

In these tests, Kir4.1 was set at 0.4 mS cm⁻², with no other conductance; corrected accordingly.

Page 10 line 269 - "...and the modelled nano-shapes (Fig. 2f..." – seems like a reference to the wrong figure.

Clarified.

Page 13 line 381 – Voltage-dependent should be erased, as Kir4.1 are not.

Erased.

Page 14 bottom paragraph (and Supplementary Figure 7) – why was the constant rate of the potassium extrusion set at K_p=0.5 mM/ms? Is there a previous reference for that?

The value was 0.5 mA/ms, in line with Kir4.1 conductance; typo corrected.

Page 35 line 1124, Figure 4f legend – the left diagram represent current density, not current as in text.

'Current density' added.

Figure 3b – measurement unit (s) should be annotated on the Y axis.

Fraction is a dimensionless parameter.

Figure 4d – color code of different frequencies does not seem to match the colors of the lines. Also, it also might be better to start the X-axis (x=0) of the right panel at the signal initiation point, so space constant could be more intuitively extracted.

Corrected.

Figure 6d – units (μm/s) should be noted for the Y-axis. Also, does the simulated data fit the in vivo measurements?

Clarified.

Supplementary Figure 6d – There is a representation of only one case, despite color code for all three.

The three curves are indistinguishable; clarified in the legend.

Supplementary Figure 6d legend, should be 'Three' not 'Thee'.

Corrected.

Supplementary Figure 9b - units should be noted for the Y-axis

MΩ added.

Supplementary Table 1 – Do the values after the ± sign represent standard deviation or standard error?

SD in the table means Standart Deviation; note added.

Reviewer #4 (Remarks to the Author):

[...] I would also suggest that in the abstract, instead of somewhat vague, “ ...some basic relationships between free Ca²⁺ dynamics and experimental readout of fluorescent Ca indicators”, a more specific key relationship be stated here.

Abstract amended as suggested.